# Biobased Kapok Fiber Nano-Structure for Energy and Environment Application: A Critical Review

**DOI:** 10.3390/molecules27228107

**Published:** 2022-11-21

**Authors:** Abdelmoumin Yahia Zerga, Muhammad Tahir

**Affiliations:** 1School of Chemical and Energy Engineering, Universiti Teknologi Malaysia (UTM), Johor Bahru 81310, Johor, Malaysia; 2Chemical and Petroleum Engineering Department, UAE University, Al Ain P.O. Box 15551, United Arab Emirates

**Keywords:** kapok fiber, hydrogen production, adsorption, degradation, CO_2_ reduction, *Ceiba pentandra*, biochar, hollow structure

## Abstract

The increasing degradation of fossil fuels has motivated the globe to turn to green energy solutions such as biofuel in order to minimize the entire reliance on fossil fuels. Green renewable resources have grown in popularity in recent years as a result of the advancement of environmental technology solutions. Kapok fiber is a sort of cellulosic fiber derived from kapok tree seeds (*Ceiba pentandra*). Kapok Fiber, as a bio-template, offers the best alternatives to provide clean and renewable energy sources. The unique structure, good conductivity, and excellent physical properties exhibited by kapok fiber nominate it as a highly favored cocatalyst for deriving solar energy processes. This review will explore the role and recent developments of KF in energy production, including hydrogen and CO_2_ reduction. Moreover, this work summarized the potential of kapok fiber in environmental applications, including adsorption and degradation. The future contribution and concerns are highlighted in order to provide perspective on the future advancement of kapok fiber.

## 1. Introduction

Several researchers have accepted that sourcing ecologically clean, inexpensive, and more durable energy on request to replace the current fossil fuel infrastructure might address the world energy challenges [1]. Due to its cleanness, high energy intensity, and renewable qualities, hydrogen is considered a viable energy conductor, particularly for renewable electricity [2]. However, a lack of feasible hydrogen generation, storage, and infrastructural technologies limits global hydrogen energy consumption [3]. On the other hand, due to the use of sustainable electric power and the great efficiency of photocatalytic reduction, the photocatalytic reduction of CO_2_ into carbon-containing fuels is a green and successful way to handle the energy issue and global warming challenges [4,5,6]. However, recently, both unexpected and planned Oil disasters have resulted on a regular basis during transportation, manufacturing, and refining, resulting in significant negative consequences on organisms and ecosystems [7]. As a result, it is required to devise a relatively effective means of eliminating the possible oil pollution concern while also recovering the spilled oil. The component that absorbs oil, typically recognized as the most effective in removing and capturing spilled oil, is commonly used [8].

Due to its distinctive physical properties, such as low density, high porosity, and specific surface area, kapok, a hollow natural fiber with multiple pores in the cell wall, has gotten a lot of interest in several study domains [9]. As a result, KF can be transformed into a greater adsorbent material capable of absorbing contaminants [10]. Once combined with a tubular construction [11], KF can also be regarded as a renewable bio-template for directing the in-situ development or coating of active elements within or outside the fiber surfaces [12] or as a raw material for other carbon-based products via pyrolysis or hydrothermal processes [13,14]. As a result, KF-based materials are already being used in a variety of applications. In addition, significant advancement has been achieved for KF-based active substances. Still, the functions of these substances in many activities, particularly in the environment and energy, have been largely undiscovered by the majority of scientists. Few review papers have been conducted on kapok fiber in recent years; for example, the study by R. Avinash Pai et al., 2015 [15] highlighted its morphology and mechanical properties, research by Yian zheng et al., 2015 [16], which focused on the application of kapok fiber as an adsorbent, and the review by Soumya Ranjan et al., 2019 [17] summarized the kapok Fiber structures and application as an adsorbent. However, there is no review paper that can reflect the current development in KF and their derived biochar materials for energy and environmental applications.

The current review paper is founded on a comprehensive evaluation of the literature using keywords such as photocatalysis, hydrogen generation, kapok fiber, adsorption, CO_2_ reduction, carbonization and pretreatment of kapok fiber, and photocatalytic degradation. This study aims to outline current advancements in kapok fiber for photocatalytic H_2_ production and CO_2_ reduction, as well as recent methodologies used for modifying kapok fiber for optimal photoactivity. To begin, this review quickly emphasizes the importance of sustainable materials and the basics, as well as the pre-treatment procedure of kapok fiber, with a particularly detailed description. Secondly, the role of kapok fiber as a substrate for the adsorption of dye, metals, and oil has been disclosed. On the other hand, the performance of kapok fiber for the degradation of heavy metals and several other contaminants is included. Finally, the most important part and the future recommendation are based on the application of kapok fiber as a co-catalyst with different semiconductors to produce energy, including hydrogen evolution and CO_2_ reduction has been discussed.

## 2. Biochar as Sustainable Material

Climate instability and environmental degradation are two of humanity’s most severe global concerns. These are inextricably tied to the growing need for economically feasible and ecologically favorable energy sources that will surely contribute to the creation of a renewable environment. Energy problems, environmental degradation, and global warming are all major issues that concern people all around the world. Consequently, a wide spectrum of people is motivated to discover simple, environmentally friendly, and cost-effective solutions to these challenges. One key part of research is the development of a variety of active materials that may be utilized to address many of the issues connected with both present and future approaches. Materials having catalytic capabilities, for example, can be produced to transform sustainable energies. Biomass may be transformed into biofuels and used as renewable energy [18,19,20]. High-capacity materials can be developed to preserve low-cost, clean, renewable solar, wind, and biomass energy [21]. Adsorbent or catalytic materials that eliminate pollutants or trap CO_2_ can be produced to solve environmental pollution and global warming challenges. Because of their capability to be used in energy storage, catalysis, adsorption, and gas segregation and collection [22,23,24,25]. Carbon materials are great alternatives for tackling numerous practical challenges. Multiple procedures, such as chemical vapor deposition, arc discharge synthesis, and calcination of synthetic or natural polymers, have been mentioned for synthesizing crystalline carbon nanotubes/nanofibers and graphene, as well as amorphous carbon, activated carbon, and carbon black substances with adjustable characteristics and functionalities. However, these approaches often need time-consuming synthesis procedures, organic solvents, and electrochemical remediation [26,27,28,29]. Biochar is described as a carbon-rich, porous solid formed by biomass pyrolysis at low temperatures (350–700 °C) in a reactor with little or no accessible air. In more scientific aspects, biochar is created by biomass pyrolysis at reduced temperatures with a restricted source of O_2_. This method frequently resembles the manufacturing of charcoal, one of humanity’s oldest industrial processes. Biochar is classified as a form of biocarbon because of its usage as a chemical source. This is characterized as a diverse spectrum of carbon compounds obtained from a variety of biological components, including plant, animal, and microbial resources. Combustion, pyrolysis, hydrothermal treatment, and gasification are some of the procedures used to create it [30,31,32,33,34]. On the other hand, biochar is produced by the combustion of lignocellulosic plant biomass or its derivatives, implying that biochar has a restricted use than biocarbon. This definition separates biochar from hydrochar, which is produced by the hydrothermal carbonization of biomass or biomass-derived organic substances such as carbohydrates and lignin at generally moderate temperatures (130–250 °C) and considerable pressure (0.3–4.0 MPa) [35]. Biochar has comparatively low porosity and surface area, but it includes rich surface functional groups as well as minerals such as N, P, S, Ca, Mg, and K, according to physical and chemical characterization using a variety of methods. Because of these qualities, biochar can be used as an adsorbent, catalyst, and catalyst support [36,37,38]. More significantly, the large-scale deployment of biochar-based active materials is seen as an ecological approach since waste biomass may be transformed into biochar, which reduces anthropogenic CO_2_ emissions. Figure 1 summarizes the sustainable theories of biochar production and uses. Biochar derived from biomass pyrolysis may be traced back thousands of years. Slow pyrolysis, quick pyrolysis, flash pyrolysis, and pyrolytic gasification are some of the pyrolysis systems that may be used to make biochar. Pyrolysis for synthesizing biochar from biomass is both cost-effective and ecologically beneficial because considerable amounts of regenerated bioenergy, bio-oil, and gas are generated parallel to the biochar itself [39].

## 3. Overview and Properties of Kapok Fiber

### 3.1. Overview of Kapok Fiber

Kapok fiber is a renewable grain fiber obtained from the fruits of the Kapok tree. It has a bright yellowish-brown color, is light in mass, fluffy, and highly hydrophobic [41]. It has an ideal fiber length of 5–20 mm and a wall thickness of 0.5–2.00 m [42,43]. The chemical content of fibers and their shape, microfibrillar angle, cell diameters, and flaws are essential elements determining their general qualities [44]. Kapok fiber is strongly acetylated and has high cellulose content. The proportion of cellulose differs depending on the area, the age of the fruit utilized, and the method of preparation employed [16,45]. Kapok (*Ceiba pentandra*) fiber is among the lightest natural fibers, eight times softer than cotton. However, kapok fibers have a waxy element on their interface, which may impact the surface attachment of the fiber substrate. As a result, these fibers are dewaxed to remove wax and other contaminants from the interface before using the mixture. Dewaxing fibers is commonly accomplished by swirling the fibers in solvents such as toluene or ethanol, followed by rinsing the fibers with distilled water. Pure kapok fiber has a crystalline structure of 54%, while dewaxed fiber has a crystallinity of approximately 63%. On the other hand, dewaxed kapok fibers have a higher proportion of crystallization due to the consistent organization achieved by removing wax and other contaminants. Several studies have previously demonstrated the utilization of kapok fibers as bio templates for multiple components, including titania (TiO_2_), nickel-cobalt (Ni-Co) layered hydroxide, and polyaniline (PANI) [10,46]. Furthermore, kapok fibers can be converted to tissue for usage in antimicrobial and chromic substances [47,48]. Kapok trees are members of the Bombacaceae family and can be found in Asia, Africa, and South America. Kapok is a soft fiber that completely covers the seeds of kapok trees (*Ceiba Pentandra*), as shown in Figure 2a–c. SEM scans of kapok Fiber cross-sections reveal that these fibers are porous in composition are shown in Figure 2d–f [49]. Kapok fiber-based products have offered up prospects for a variety of novel utilization domains due to their particular properties. As a result, kapok fiber and related mixed components have received more interest in recent years. Because of its exceptional buoyancy [50], Kapok fiber has long been applied as a filler in bed sheets, upholstery, life protectors, and other water-safety systems, as well as for thermal and sound isolation due to its air-filled porosity [51]. Due to its temperature preservation qualities, KF may be mixed with different fibers to create garment materials with specific attributes. Kapok fiber, a lignocellulosic plant fiber, has been employed as a reinforcement ingredient in polyester composites through hybridization with glass and sisal fabrics [52,53]. In addition, KF can be mixed with thermoplastic cassava starch (TPCS) to minimize water absorption and increase the tension at the highest loading and Young’s modulus of the TPCS/kapok fiber compound [54]. Because of the wax coating on its top-layer, this fiber has good hydrophobic-oleophilic properties, and as a result, it is gaining popularity as an oil absorption substance as well as in packaging paper that requires durability and waterproofing [8,55,56,57,58,59,60,61,62]. Based on the native hollow form, kapok fiber can also be used as a suitable model substance or support material, such as a catalyst transporter [63]. Furthermore, this fiber is a viable starting source for producing flexible activated carbon fibers or a second-generation bioethanol generator [64,65]. The simultaneous hydrothermal development of C-doped g-C_3_N_4_ and TiO_2_ in a single pot utilizing kapok fiber as a bio-template is a straightforward, ecologically friendly, cost-effective synthesis technique.

Furthermore, this section has highlighted and summarized the several properties of kapok fiber and its different uses, which qualify it as a great sustainable material for energy production and other environmental applications.

### 3.2. Kapok Fiber Composition

Kapok fiber is an organic grain fiber that is highly lignified and mostly composed of cellulose, lignin, and xylan. According to several sources, the chemical content of kapok fiber differs. Another research discovered that kapok fiber biologically consisted of 13% lignin, 23% pentosan and 64% cellulose by mass. In contrast, another discovered that kapok fiber was composed of 21.5% lignin, 22% xylan and 35% cellulose, with an elevated ratio of guaiacyl/syringyl units (6–4) and a strong amount of acetyl groups (14.0%) when compared to ordinary fiber (Table 1) [16]. The variations could be attributed to variances in kapok origins and preparation procedures. However, given its colossal lumen, the crystallization level of kapok fiber has been estimated to be 35.90%, the specific refraction is 0.017, and the bulk density is 0.30 g/cm^3^. Kapok fiber comprises two primary layers, each with a different microfibrillar structure. The outer surface includes cellulose microfibrils positioned longitudinally to the fiber axis. In contrast, the inner layer contains fibrils almost perpendicular to the fiber axis [68]. Optical microscopy and scanning electron microscopy revealed that kapok fiber has a tubular form, a smooth surface, and thickening grayness at the ends. Kapok fiber has a smooth texture with a dense coating of wax, and the cross-section is rectangular to circular with a wide tubule and thin wall [69]. This porous structure distinguishes KF from other fibers and gives it a porosity of approximately 80% [51]. Cotton fiber, another unicellular fiber, has a compacted ribbon-like shape and rolls in a spiral pattern along the center [69]. Raw kapok fiber cannot be woven as cotton fiber due to its fragility, low cohesivity, and hardness, but it can be effectively combined with cotton fiber to make yarns. In addition, increased kapok proportion reduces fiber consistency and toughness while increasing yarn flexibility, resulting in a lower overall cost of manufacturing for the fibers [70]. Furthermore, the big aperture and waxy surface are not conducive to the entry of aqueous watercolor additives or pigments, so kapok fiber has poorer dyeing effectiveness [71]. The mixture of these characteristics gives kapok fiber improved hydrophobic-oleophilic qualities, allowing it to be employed as a buoyant substance as well as an oil-absorbing substance.

## 4. Preparation and Treatment of Kapok Fiber and Derived Materials

Surface treatment of kapok fiber can be used to improve the natural properties or change the surface qualities. Surface treatment can include (1) chemical treatment, such as alkali/acid treatment, solvent treatment, oxidation treatment, acetyl treatment, and (2) physical treatment, such as ultrasonic treatment and radiation treatment. The surface qualities of kapok fiber can be enhanced through suitable treatment (Figure 3), which causes alterations in physical and chemical processes at the contact.

### 4.1. Chemical Treatment

Chemical treatment can form persistent ionic bridges across the interface of the fiber and a layer or generate additional receptors based on the form and properties of the responding component. Alkali treatment of fibers is considered the best famous and effective ways of chemical treatment, and it has been applied successfully to clean practically all-natural fibers [44]. Initially, alkali pretreatment of kapok fiber was applied to eliminate lignin, pectin, wax, and natural oils from the fiber’s outer surface, enhancing its surface and physical qualities for polymer usage (Figure 4b) [52,72]. The primary frequently applied reagent for removing and/or flushing kapok fiber surfaces is sodium hydroxide. Alkali pretreatment has no discernible impact on the chemical constitution of cellulose. Still, it increases the proportion of amorphous cellulose at the expense of crystallized cellulose, altering the more delicate texture of natural cellulose I to cellulose II through a mechanism defined as alkalization [73,74].

This approach is complemented by considerable kapok fiber de-esterification, leading to a significant decrease in wave frequency at 1740 and 1245 cm^−1^ in the infrared spectrum for NaOH-treated KF relative to pure kapok fiber [8]. Additionally, the alkalization of kapok fiber alters its surface texture by demonstrating a rough surface morphology (Figure 4d). Tiny fibrils, cracked holes, and superficial gaps can be noticed on the exterior of NaOH-treated kapok fiber. In contrast, pure kapok fiber has a soft texture Figure 4a,c, revealing the reduction of the removal of external wax from kapok fiber results in the sensitization of hydrophilic hydroxyl radicals is favorable in fiber-matrix interactions. Attachment due to the support of both physical interconnecting and adhesion interactions to chemicals like resins and dyes. However, the elimination of wax and the implosion of the solid, porous form may diminish the oil sorption potential. Data reveal that after 8 h of alkali pretreatment, a sample oil’s sorption rate is reduced by 26.3% compared to pure kapok fiber [61].

The acid procedure can also eliminate plant wax from kapok fiber, as evidenced by relative FT-IR findings demonstrating modifications in the absorption wavelengths at 3410 and 2914 cm^−1^ because of the exposure of cellulose hydroxyl radicals in fiber barriers; however, no significant reduction in other absorption wavelengths. Utilizing toluene, chloroform, n-hexane, and xylene as template oils, HCl-treated KF demonstrates excellent oil absorbency over NaOH-treated [8]. Tye et al.’s 2012 study has proved that kapok fiber can only generate 0.8% lowering sucrose by enzymatic hydrolysis without treatment. However, acid pretreatment can improve the output of decreasing sugar to 85.2%. Nonetheless, alkaline processing is more successful in generating a greater decreasing sucrose rate. This is because alkali therapy affects the cellulosic parts of the kapok fiber and the no cellulosic parts (hemicellulose, lignin, and pectin). In contrast, acid treatment can eliminate just hemicellulose, particularly xylose, from pure KF (Figure 5A,B). Overall, plant fiber with a minimal lignin level is better adapted for enzymatic hydrolysis because lignin adsorbs enzymes and, as a result, reduces enzyme performance, as Figure 5C shows the size dispersion of adsorbent particles as assessed by dynamic light scattering devices. Based on the Eun Ae Park et al. study, the dehydrated kapok was washed with a 36% HCl solvent to eliminate impurities and accelerate hydrolysis (Figure 5D,E). Figure 5F shows a detailed view of zeolite fragments occluded by an arrangement of HCl-treated cellulose fibers. It was subsequently processed with gaseous HCl to improve incineration performance in order to create an HCl-treated kapok composite (HKC) membrane, as illustrated in Figure 5G [76].

As a result, alkali treatment is more successful in increasing cellulose transformation from kapok fiber to glucose to manufacture second-generation bioethanol [65]. The two most commonly used solvents for the processing of kapok fiber are chloroform and ethanol. Because of its hydrophobicity, which is similar to the waxy surface of kapok fiber, chloroform was selected as one of the solvents. For chloroform-treated kapok fiber, FT-IR spectra reveal a rise in absorption wavelengths at 3410 and 2914 cm^−1^ but no apparent change in other absorption spectra, implying simply wax reduction from the kapok surface [8,61]. The gap between 4 and 8 h of extraction is not substantial, with the latter achieving a marginally higher percentage of wax reduction. On the other hand, Solvent-treated kapok fiber is predicted to have lower oil absorbency as an oil-absorbing substance. After 8 h of chloroform treatment, there is a 2.1% drop in sorption ability related to pure kapok fiber, but no significant morphological variation is reported [61]. According to the appearances, after removing the surface wax, the chemical treatment of kapok fiber diminished its smooth shine. Furthermore, chemical treatment degrades kapok fiber’s hydrophobic-oleophilic characteristics; for example, premature advancements have been noticed throughout the deep-bed purification method while utilizing ethanol- or chloroform-treated kapok fiber as the filter substrate [77]. Furthermore, to chloroform and ethanol, a 1:2 combination of alcohol and benzene is utilized for kapok fiber pretreatment to acquire dewaxed kapok fiber [78].

An oxidation process with NaClO_2_ is often used to transform the hydrophobicity of kapok fiber to hydrophilicity. This technique may eliminate certain phenolic substances from kapok fiber, reducing the lignin percentage of kapok fiber from 20.9% to 2.6% [79]. This finding is supported by a drop in total alkaline nitrobenzene oxidation (NBO) output from 78.4 mg/g CWR (cell wall residues) for a control sample to 10.5 mg/g CWR for a NaClO_2_-treated sample [69]. According to FT-IR spectra, the absorption peaks between 1602 and 1504 cm^−1^ practically vanished in NaClO_2_-treated kapok fiber, indicating the breakage of the aromatic chain in lignin [8]. The NaClO_2_ process is an efficient method for changing the surface character of kapok fiber. According to research, a watered decrease has a significant contact angle (θ = 116°) on raw kapok fiber. In contrast, water drops sink swiftly into kapok fiber treated with NaClO_2_, forming a vast distributing circle on the surface [55]. Furthermore, NaClO_2_ treatment changes the aggregation form and expands the porous arrangement in kapok fiber, with the crystallinity index decreasing from 35.34% to 26.97% for untreated and NaClO_2_-treated kapok fiber, correspondingly. Following that, NaClO_2_-treated kapok fiber absorbs more oil, with rate improvements of 19.8%, 30.0%, 21.5%, and 24.1% for toluene, chloroform, n-hexane, and xylene, correspondingly. Furthermore, NaClO_2_-treated kapok fiber is more reusable, implying a high capacity for oil remediation. Because of the hydrophilic properties imparted by decolorization, NaClO_2_-treated kapok fiber may be (i) effectively tailored to absorb the cationic dye methylene blue (MB) from an aqueous solution, with an adsorption rate of 110.13 mg/g [80]; (ii) attached to glycidyl methacrylate (GMA) while being irradiated with Co-60 gamma rays so that the consequent grafting material has additional functionalities such as ion exchange and adsorption qualities while retaining the initial characteristics [79]. After several oxidation processes, such as NaClO_2_–NaIO_4_–NaClO_2_, the generated oxidized kapok fiber can be employed as an effective heavy metal adsorbent. The formation of –COOH groups is responsible for the increased adsorption of heavy metal ions onto chemically oxidized kapok fiber [69]. In comparison to oxidation, acetylation is a highly appealing approach for changing the surface of kapok fiber to make it very hydrophobic. The method’s basic premise is to combine the fiber’s hydroxyl groups (OH) with acetyl groups (CH_3_CO) since the acetyl group is more hydrophobic than the hydroxyl group [44]. Kapok fiber can be acetylated using or without a catalyst to attach acetyl groups onto the cellulosic matrix. According to FT-IR research, the absorption levels at roughly 1742 cm^−1^, 1375 cm^−1^, 1244 cm^−1^, and 2910 cm^−1^ rise following acetylation [56]. Additionally, when acetic anhydride increases, the crystallite value decreases from 34.34% for raw fiber to 22.90–29.42% for acetylated fiber. Therefore, an increased percentage of acetic anhydride produces a reduction in the cumulative form of lignocellulose [81]. As a result, effective acetylation is projected to optimize the efficiency of kapok fiber compounds by supporting improved fiber-to-resin bonding and the oil sorption potential due to increased surface quality [56]. To eliminate waxes found on vegetation membranes (Figure 6), according to Qiang Zhang et al., the KF was extracted using a Soxhlet device with a 2:1 solution of toluene/ethanol for 6 h at 115 °C. Following that, 3 g of KF was placed in a 250 mL (1%) and 0.5 mL (36%) blended solution and responded at 75 °C for 4 h until the yellow coloration vanished. The reacting fibers are then rinsed with deionized water to neutralize them. The KF was then placed in a 250 mL (5 wt%) KOH solution for 12 h [82].

The oil absorption rate of kapok fiber treated with water, HCl, NaOH, NaClO_2_, and chloroform was examined in this work. SEM characterization revealed the alteration in surface shape, chemical compositions, crystallinity, and surface elemental composition following various treatments. The removal of plant wax had no effect on the oil sorption ability of kapok fiber, according to the findings. However, the NaClO_2_ treatment improved the oil absorption rate of the kapok fiber structure. Furthermore, following compressing, the fiber retained the majority of its oil sorption capacity. Based on the high oil absorption, outstanding reusability, and superior biodegradability, treated kapok fiber is a viable alternative to conventional synthetic oil sorbents used in oil extraction in the absence of water. Globally, the oxidation method is considered the more significant process to treat kapok fiber due to its several advantages, including changing the aggregation form, expanding the pores, and absorbing more oil and dye. Comparing to the alkaline, the acid and the chloroform procedure defect the KF structure, reducing enzyme performance and offering a simple wax reduction, respectively.

### 4.2. Physical Treatment

Figure 7 depicts SEM micrographs of raw kapok fibers, treated kapok fibers, and derived cellulose. Pure kapok fiber has a silky look and a smooth surface (Figure 7a,b). The general tubular form remained intact. After alkali processing, the air entrapment within the kapok fiber was removed (Figure 7c–f). The structure was totally flattened, resulting in a flat ribbon-like shape. In addition, the kapok fibers surface was rougher than the untreated fibers. Alkali treatment has been shown to promote crystallinity by removing non-cellulosic particles. The SEM illustration of the produced cellulose after chemical treatment with NaOH and NaClO_2_ is shown in Figure 7c–f. The diameter variation after treatment with NaOH and NaClO_2_ was 4.5 to 8.5 m, whereas, it was 18.5 to 23.1 m for samples treated exclusively with NaOH [84].

Based on the inherent benefits, kapok fiber is gaining popularity as an eco-friendly material. Despite growing interest in kapok fiber, the difficulties in spinning it restrict its manufacturing potential for larger uses. This may be partly addressed by mixing kapok fiber with other polymer matrices to create various composite materials. To increase the physical and chemical activities at the interface and obtain better efficiency in composite materials, kapok fiber should be processed chemically or physically to improve intrinsic qualities or change surface features. With a focus on this green cellulosic fiber, additional research should be conducted to broaden the application domains for kapok fiber by leveraging its combination of high hollowness and hydrophobic-oleophilic properties.

### 4.3. Carbonization Process

The combustion of biomass has recently gained popularity due to the adaptability of carbonized resources for prospective usage in a variety of industries [85]. Overall, activated carbon fiber (ACFs) from KF could be produced through a simple combustion technique in oxygen-free air. The following is an example of a standard method: KF was carbonized for 1 h under a nitrogen atmosphere at 973.15 K, then activated by steam for 50 min at 973.15 K, and finally neutralized by cleaning in hot water and vacuum drying for 24 h at 393.15 K. The S-BET of KF-based ACFs was 1510.0 m^2^/g, significantly more significant than the native fiber [86]. As a result, the KF-based ACFs could function as one type of effective adsorbent for eliminating various dyes [87]. The carbonized kapok filament could also serve as a carbon electrocatalyst for CO_2_ electroreduction. It has been observed that carbonized KF has the capacity to effectively transform CO_2_ to format without the need for active elements to be doped. Furthermore, carbonized KF was proposed as novel catalyst support for in-situ attaching metals (Sn, Bi, Pb, and Cd) on KF tubes for further electrocatalytic CO_2_ removal [88]. Utilizing KF as a precursor component, Song et al. (2020) created a light, hydrophobic, and high porosity carbon microtube aerogel (CMA) capable of absorbing oils at a rate of 78–348 g/g. Pyrolysis KF, in particular, is important in electrode materials for supercapacitors since a distinctive hollow form formed from KF can increase internal electrolyte transit by lowering dispersion tolerance [89].

Based on Rui Jun Wang et al.’s 2018 method, the KF was left to dry for 48 h at 60 °C before being processed with boiled NaOH solution for 50 min to eliminate the wax layer [90]. After that, deionized water was used to eliminate the NaOH until the residue was balanced. Following that, the pretreated KF was dehydrated at 60 °C for 24 h. PKF was frequently poured into a 20 mL porcelain crucible before being put in a 100 mL porcelain crucible. The porcelain crucible was placed in a box-type pressure furnace and pyrolyzed to four several temperatures, 700, 750, 800, and 900 °C, with a heating rate of 5 °C min^−1^ and a residence time of 2 h (Figure 8). Following pyrolysis, the resultant charcoal substance was bathed in an additional HCl aqueous solution to eliminate ash before being separated and rinsed with deionized water until the filtrate became neutralized. The carbon materials were dried at 60 °C for 24 h and labeled as CKF-700, CKF-750, CKF-800, and CKF-900. At a present density of 1 A g^−1^, the CKF-750 biochar has the highest specific inductance value of 283 F g^−1^ at a combustion temperature of 750 °C [91,92,93]. Furthermore, the biochar exhibits excellent cycle consistency and rate performance, indicating that kapok fiber is an appropriate feedstock for producing biochar [94], which is employed as an electrocatalyst in supercapacitors [90].

FE-SEM was used to examine the microstructures of KF and biochar samples CKF-700, CKF-750, CKF-800, and CKF-900 (Figure 9). KF (Figure 9a,b) demonstrates a faultless hollow tubular design with an outside diameter of 17.4 m, an interior diameter of 16.4 m, and a wall thickness of 0.5 m. A deeper view indicates that KF has a smooth texture and a layered arrangement. It has been revealed that the kapok fiber wall has 5 structures [95]. CKF-700 (Figure 9c,d) has a highly smooth surface, even softer than natural KF, showing that 700 °C is too low for KF to be pyrolyzed to a significant extent. With detailed examination, several nanoscale pits are predecessors of micropores, and they are all caused by the pyrolysis of certain membrane active groups in Kapok Fiber [96]. When the temperature is raised to 750 °C, the CKF-750 exhibits a rough texture and micropores all over the membranes (Figure 9e,f). This also demonstrates that a higher temperature of 750 °C causes KF to be pyrolyzed considerably, resulting in the appearance of micropores in CKF-750. The higher-intensity SEM scan in Figure 9e shows that CKF-750 does have multiple micropores. FE-SEM images of CKF-800 (Figure 9g,h) and CKF-900 (Figure 9i,j) reveal rough surfaces but clearly smaller micropores than CKF-750. This could be owing to the convergence of micropores caused by the high temperature. These findings imply that the pyrolysis temperature has a significant impact on the microstructure of biochar [97], particularly the pore shape. The calcination temperature of 750 °C is ideal for capillary generation. A low temperature is detrimental to the growth of apertures, whereas a high temperature causes cavities to combine [98]. The cylindrical morphology of the four biochar particles varied as the pyrolysis temperature increased. At calcination temperatures of 700 °C, the CKF-700 biochar retains a more coherent porous form with only minor debris spread on its top. However, when the calcination temperatures rise 700 °C, the as-prepared biochar samples struggle to maintain their entire vacuous shapes. CKF-750 has a curved surface, a deflated porous shape, and several thinner sheets, which increases the specific surface area of CKF-750. As previously stated, kapok fibers have five layers, and greater temperatures generate tension among the layers, causing the layers to form sheets. The features of CKF-800 and CKF-900 biochars are identical to those of CKF-750, except that excessively high temperatures cause the micro-pore structures to combine simultaneously.

The thermogravimetric examination was conducted to verify the appropriate heating value, and the DTG and TGA graphs of KF are presented in Figure 9k. There was a minor mass reduction level below 100 °C, which could be attributed to the evaporation of water atoms in pure KF, comprising absorbed and infiltrated water molecules. The breakdown of hemicellulose generated a 10% mass decrease between 100–230 °C [99]. Severe mass reduction of up to 63% occurred from 230 to 380 °C, caused by cellulose destruction [100]. The following curves exhibit a stable tendency until around 770 °C. Mass reduction may occur over 770 °C due to the restructuring of macromolecular aromatic compounds. This finding is consistent with the published kapok fiber content [54]. Overall, pyrolysis KF produced at high carbonization temperatures has excellent conductance and an abundance of microstructure, which makes it an excellent conductor substance. KF samples generated at moderate calcination temperatures, on the other hand, typically exhibit unsatisfactory electrochemical efficiency. In contrast, at combustion temperatures of 500 °C and 600 °C (Figure 9), the resulting kapok fiber samples have limited particular impedance and substantial voltage dips (IR drops), indicating high levels of resistance. As a result, four alternative calcination temperatures were used to create the KF: 700 °C, 750 °C, 800 °C, and 900 °C. The FT-IR bands can provide precise data on the surface functional groups of KF (Figure 9l) and calcinated KF samples (Figure 9m). The absorption band at 3402 cm^−1^ in Figure 9l is attributed to the O-H stretch wave. C-H stretching wave of the alkyl is ascribed to the wavelength at 2919.7 cm^−1^. The absorption band at 1737.6 cm^−1^ is in the region of the C=O bending vibrations of the cellulose ester band. The C-O is responsible for the band at 1054.9 cm^−1^. At roughly 1504.2 cm^−1^, there are many contaminated peaks related to benzene skeleton vibration [101]. The FT-IR spectrum of the C-KF materials is shown in Figure 9m. As the carbonization frequency increases, the wavelengths of the carbonaceous materials grow smoother and simpler, implying that most of the functional groups on the carbonaceous materials are degraded. The FT-IR spectrum of pure kapok fiber show O-H (3442 cm^−1^) and C-O (1083 cm^−1^) peaks. The presence of benzene compounds is suggested by the absorption region at 1592 cm^−1^. The functional groups in all of the C-KF samples are comparable, but there may be a slight variance in the number of functional groups. However, the crystalline compositions of KF and C-KF samples were investigated. The XRD profile of KF (Figure 9n) displays two peaks at 2θ = 15.81° and 22.58°, which correlate to the cellulose crystallography planes (101) and (002) correspondingly [102]. The peak at 2 = 22.58° is thin and peaked, reflecting that the crystallographic particle length of cellulose is greater and the crystallization is stronger. There are two distinct spectra in the pyrolysis samples (Figure 9o) at 2 = 23.28° and 43.77°, which correlate to the (002) and (100) crystallography planes of carbon, correspondingly [103]. Those small and expansive peaks obviously imply decreased crystallographic particle sizes, reduced crystallinity, and amorphous pyrolysis KF forms.

The Raman band of the CKF compounds is shown in Figure 9p. There are two distinct points, 1341 cm^−1^ and 1585 cm^−1^. Irregularities and instability cause the peak at 1341 cm^−1^ (D), whereas the peak at 1585 cm^−1^ (G) is caused by graphitic composition. The carbonization level is represented by the difference between two peak intensities (ID/IG). The ID/IG ratios of CKF-700, CKF-750, CKF-800, and CKF-900, correspondingly, were 0.95, 1.03, 1.01, and 1.02, indicating that the CKF materials have a lesser level of carbonization. In other words, the as-prepared materials had a low level of carbonization and were unstructured [104].

Based on the distinctive thin-walled tubular shape, KF has subsequently gained more interest in energy-related applications. Simple hydrothermal carbonization or pyrolysis carbonization is frequently used to unlock its full potential as an electrode material for supercapacitors. Essentially, these studies are also restricted to the laboratory, and when paired with the energy demand of the carbonization process, the practical application still needs to be revised. Further research into genuine energy concerns should be prioritized, with real solutions in consideration.

### 4.4. Coated of Kapok Fiber

A simple surfactant-assisted approach was used to create polyacrylonitrile (PAN)-coated vacuous kapok microtubes. CTAB (cetyltrimethylammonium bromide) was used to aid polymer formation and organization. After PAN coating, the kapok fibers maintained their hollow microtube form. The water interaction point was also reduced dramatically from 133.43 to 0°. At greater CTAB concentrations, the PAN coating became denser and more homogeneous. A higher level of AN, on the other hand, occurred in a denser PAN layer. Serial studies were used to explore the influence of interaction duration and the effect of temperature on the methyl orange dye removal and copper out of an aqueous solution. Isotherm investigations indicate that MO and Copper (II) elimination is based on the Langmuir isotherm concept, with maximal sorption capacities of 34.72 and 90.09 mg/g, correspondingly. Copper (II) is selectively immobilized on PAN-KF above MO, according to the findings. Kinetic investigations show that methyl orange and Cu (II) immobilization on PAN-KF is based on the pseudo-second-order kinetic concept, with chemisorption as the rate-determining mechanism. Furthermore, respectively Copper (II) and MO immobilization thermodynamic records are endothermic and random [105].

The SEM scans of KF after PAN application with considerable CTAB values (10–30 mg) are shown in Figure 10a–d. While approximately 10 mg of CTAB was employed, massive agglomerates of PAN fragments were minimally attached to the kapok layer, as shown in Figure 10a. PAN compounds range in dimension from 4 to 12 m. The coating appeared to be fragile, as it could be quickly peeled from the fiber interface by light erosion. However, a rise in CTAB dosage from 10 to 15 mg resulted in a significant enhancement in coating persistence and a decrease in PAN fragment diameter to approximately 2 m. As shown in Figure 10b, the PAN surface is substantially softer. Furthermore, the level of PAN aggregation reduced dramatically across the fiber interface. Raising the CTAB concentration to 20 mg yielded a uniform covering of PAN on the kapok surface. As demonstrated in the inset of Figure 10d, there are no PAN accumulations. The kapok fiber combination has outer and inner lengths of approximately 27.63 and 22.97 m, correspondingly. Data suggests that the PAN layer is approximately 0.67 m of thickness. The kapok fibers kept their porous form in all CTAB doses tested, indicating that the top kapok layer area was not reduced. These findings definitely suggest that CTAB has a substantial role in distributing PAN atoms throughout the kapok fiber top layer [106].

The related FTIR bands of pure KF and PAN-KF mixtures treated with 10–30 mg of CTAB are shown in Figure 10e. The FTIR wavelengths of pure KF and PAN loading KF revealed numerous distinct regions. In both pure and PAN loading KF, significant peaks owing to C-H stretching, C-H bending, and C-C stretching are found around 2900, 1200–1400, and 1020 cm^−1^, correspondingly. For cellulosic substances, all bands are frequent. The broad spectrum of pure kapok at around 3400 cm^−1^ resulting from the stretching resonance of O-H in cellulose, on the other hand, has become less noticeable following PAN coating in all CTAB concentrations tested. Furthermore, PAN’s composition is devoid of OH functional groups. The FTIR spectrum for PAN-kapok fibers, in Figure 10e, shows significant peaks of C≡N extending at roughly 2250 cm^−1^, a nitrile chemical component missing in pure kapok material. This peak corresponds to the top common active group in PAN and indicates the effective loading of PAN on kapok fibers. CTAB may aid in the adhesion of the negative charge acrylonitrile (AN) monomer by initially forming a hydrophobic contact with the fiber interface, with their positively charged heads pointing outwards. Electrostatic contact allows the negatively charged AN monomer to bind and coordinate on the substrate. Following additional polymerization, a thin hydrophilic PAN layer is formed [105]. The impact of interaction duration on the immobilization performance of PAN-kapok hollow micro-tubes for Copper (II) and MO from an aqueous media is shown in Figure 10f. There was a quick Cu (II) elimination throughout the first phases, and equilibrium was reached in 120 min. In contrast, progressive growth in MO was observed for the first 100 min, with stability being reached after 150 min. Both the Cu (II) and MO adsorption processes are thought to be relatively quick.

Although raw fiber is compared to surface-coated fiber, the rough surface with low surface energy efficiently prevents oil from flowing from the fiber structure, leading to a high oil sorption rate. This innovative oil sorbent can collect oil in aquatic settings in a simple, rapid, and highly effective approach. It is predicted to replace typical synthetic fiber for large-scale cleaning of spilled oil from water surfaces.

## 5. Applications of Kapok Fiber

Photocatalytic Kapok fiber is recognized as one of the most resilient and effective material sources accessible for environmental rehabilitation and energy production due to its exceptional photocatalytic performance, hollow structure, great renewability, and compressibility. There are, however, few detailed reviews on this matter with strong photocatalytic activity. Therefore, the most recent explosive advancement in photocatalytic kapok fiber, including various kapok fiber materials and overall fabrication methodologies, was examined to evaluate this advanced research. Pollutant absorption, photocatalytic degradation, hydrogen production, and CO_2_ reduction were the main applications of this photocatalytic Kapok fiber (Figure 11).

### 5.1. Photocatalytic Hydrogen Production

Compared to various organic fuels, hydrogen is designated as a sustainable solar energy source, with a thermal efficiency of 120–142 MJ kg^−1^. Currently, the global generation of hydrogen exceeds 44.5 million tons [3], and it will be the primary generator of power until 2080 [107]. However, there are numerous methods for producing hydrogen-like substances, and the most common is thermolysis [108], electrolysis [109], biomass [110], photolysis [109], and hydrolysis [111,112,113,114]. Photocatalytic water dissociation is a successful technology that has received a lot of interest due to its broad application for power and sustainability purposes. Artificial photosynthesis is one of the only sustainable, long-term answers to the forthcoming fuel and environmental crises [115]. A group of scientists has already investigated several photocatalyst substances [116]. Nevertheless, the majority of photocatalysts respond to ultraviolet (UV) light, which accounts for only 5% of solar energies [117]. TiO_2_ has been intensively investigated as a prospective option for hydrogen production due to its acclaimed physical and chemical qualities, high permanence, earth-abundant, low cost, and non-toxicity. However, its large bandgap (3.0–3.2 eV) limits its absorptivity range [118]. Furthermore, graphitic carbon nitride (g-C_3_N_4_) has received a great deal of attention due to its physical and chemical characteristics, low cost, earth-abundant, simple preparation, renewability, and, most importantly, visible light responsiveness due to a narrow bandgap of 2.7 eV [119,120,121]. However, the photocatalytic performance is limited by the high recombination rate and low surface area. Although several scientists have examined the heterojunction of TiO_2_ and g-C_3_N_4_ since it has a high surface area, decreases electrons recombination, and absorbs visible light. The photocatalytic performance has not progressed significantly [122].

The hydrothermal bio-template procedure was used to generate a C-doped g-C_3_N_4_ effectively using the carbonization procedure at 500 °C and nitrogen-coated titanium dioxide as a core-shell heterojunction photocatalyst Figure 12. During constructing a core-shell heterojunction photocatalyst, kapok fiber was exploited as a bio-template and in-situ carbon coated in CN and titanium dioxide. Furthermore, urea application as a g-C_3_N_4_ precursor contributes to band-gap narrowing in TiO_2_ via in-situ C and N loading. Several characterization approaches were used to investigate the impact of TiO_2_ source amount on the formation of core-shell nanocomposite heterojunction photocatalysts, which can impact and enhance catalytic performance. The photoelectrochemical and photoluminescence investigations revealed that the bio-template core-shell nanocomposite heterojunction photocatalysts had a remarkable improvement in photoinduced electron-hole dissociation performance. The improved photogenerated charge carrier dispersion and shorter band gap lead to enhanced photocatalytic activity, with the CCN/T-1.5 material producing the most hydrogen (625.5 μmol h^−1^ g^−1^) in methanol medium [123].

Figure 13a shows that at 13.1° and 27.5°, the pure bio-templated CCN exhibits two different summits. The peak at 13.1° (100) is caused by in-plane repeating tris-triazine units, whereas the peak at 27.5° is caused by interconnected aromatic layering sheets of graphitic carbon nitride (002). Overall, the positioning of these two XRD peaks of bio-templated CCN is identical to that of pure g-C_3_N_4_ but differs in terms of crystallization, as previously stated [124]. Significant changes in XRD peaks can be seen between photocatalyst samples with varied TiO_2_ precursor amounts. The contemporaneous formation of TiO_2_ nanoparticles within inter-planar stacking of conjugated aromatic sheets of g-C_3_N_4_ was revealed by the lowering of (002) peak frequency of CCN as TiO_2_ precursor proportion increased. Unexpectedly, with a titanium dioxide dosage of 1.5 M, the (002) peak amplitude of the CCN began to fade. In contrast, at 2θ = 25.4°, the little shoulder of the (101) central crystal axis of the nanocrystalline phase characteristic began to develop. Therefore, the specific peak of the nanocrystalline phase is difficult to discern because it overlaps with the distinctive peak of the CCN at 27.4–27.6° [123].

According to Zhang et al., 2019, using one-step carbonization of kapok fiber (KF), a flexible material, was combined with melamine to make carbon coating g-C_3_N_4_. The biochar ribbon edges generated following KF disintegration serve as a substrate for CCN epitaxial expansion. In addition, the photocatalytic performance in H_2_ from water splitting has been studied. After C-loading from Kapok breakdown, the biochar ribbon edges at the thin CN layers promote charge dissociation and transfer for the surface H_2_ production process. The CCN has better visible-light-driven photocatalytic activity, with an H_2_ production rate of 18.89 mol/h, 67.5 times greater than the pure CN (0.28 mol/h). The observed quantum outputs for monochromatic light = 420, = 470, and = 550 nm are determined to be 4.1%, 1.4%, and 0.66%, respectively [125]. The FTIR spectra for the CNs and CCNs samples are shown in Figure 13b. The broad vibration bands at 3000–3500 cm^−1^ are frequently attributed to O-H and N-H stretching vibrations, implying partial hydrogenation of nitrogen atoms on the margins of CN layers [126]. CN stretching vibrations in CN heterocycles are responsible for multiple resonance bands at 1000–1700 cm^−1^ [127]. Additional vibration brands are formed after CN is synthesized at greater temperatures, showing incomplete amorphism of CN, and additional active group branches are formed following higher temperature carbonization. The 814 cm^−1^ band is caused by the vibration mode of the s-triazine unit. Clearly, CCN and CN have identical FTIR bands, showing that the addition of kapok throughout melamine calcination does not affect the creation of the graphitic carbon nitride shape.

**Figure 13 molecules-27-08107-f013:**
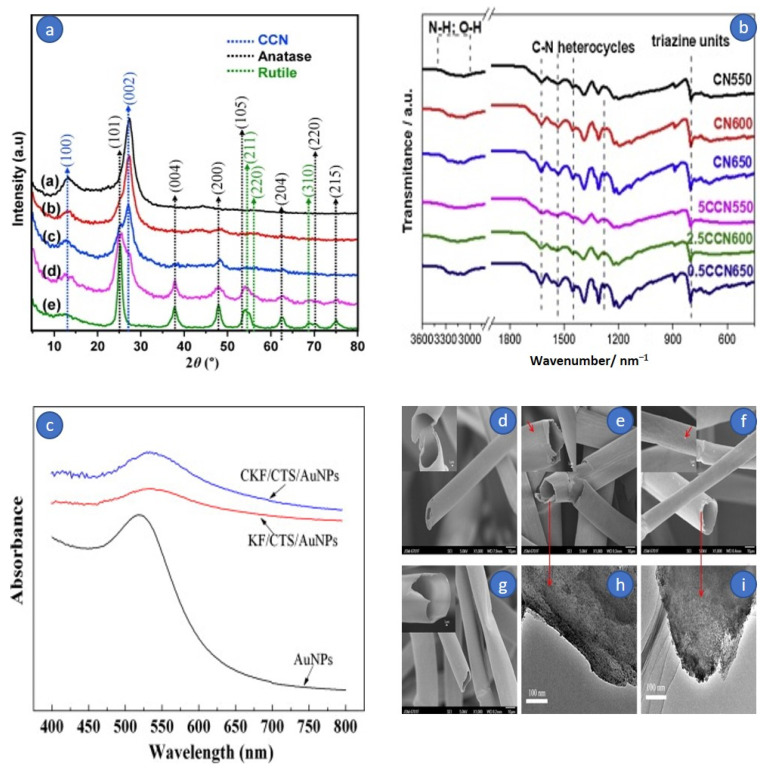
(**a**) C-doped g-C_3_N_4_/TiO_2_ heterojunction photocatalyst XRD dispersion pattern regenerated by bio-template technique with varied TiO_2_ precursor concentration. (**a**) Pristine CCN. (**b**) CCN/T-0.5. (**c**) CCN/T-1.0. (**d**) CCN/T-1.5. (**e**) CCN/T-2.0. Reprinted with permission from Ref. [123]. Copyright 2019, copyright Elsevier. (**b**) FTIR wavelengths of CN550, CN600, and CN650 samples, as well as KF-treated samples (5CCN550, 2.5CCN600, and 0.5CCN650). Reprinted with permission from Ref. [125]. Copyright 2019, copyright Elsevier; (**c**) The UV–vis bands of AuNPs, KF/CTS/AuNPs, and CKF/CTS/AuNPs; (**d**) SEM scans of KF. (**e**) CKF. (**f**) KF/CTS/AuNPs. (**g**) CKF/CTS/AuNPs, with the magnification of scans highlighted. TEM images of (**h**) KF/CTS/AuNPs and (**i**) CKF/CTS/AuNPs. Reprinted with permission from Ref. [128]. Copyright 2014, copyright Elsevier.

Based on Wang et al., 2014, the generated KF with or CKF/AuNPs/CTS nano-composites exhibit excellent catalytic performance and durability for the catalytic reduction decolorization of CR dye, with the color of the CR solution rapidly fading within Three minutes at a modest catalyst dosing of 0.3 g/L. Furthermore, after 20 min of sonication or 1 mol/L acid treatment, the nanocomposite’s catalytic performance can be sustained. The hydrogen was created concurrently with the catalytic decolorization of CR and can be recovered as a clean, renewable energy, with a total output of 430 mL/L. As a result, the nanostructure could be utilized as a catalyst to decolorize dye effluent and create H_2_ in a single cycle [128]. The absorbance peak at 533 nm was definitely observable in the UV–vis spectra of KF/CTS/AuNPs and CKF/CTS/AuNPs, confirming that the AuNPs were effectively bonded on the membrane of CTS-coated KF and CKF as predicted (Figure 13c). This finding is also clearly displayed in the test procedure, where the purple-red AuNPs solution becomes colorless when added to CTS-coated KF or CKF.

FESEM was used to examine the surface morphologies of KF, KF/AuNPs, KF/CTS/AuNPs, and CKF/CTS/AuNPs, while TEM was used to investigate the AuNPs on the membrane of Kapok (Figure 13). It is easy to see that KF has a smooth surface (Figure 13d). However, the membrane of kapok becomes moderately rougher with a minor form after carboxylation (Figure 13e), showing that chloroacetic acid interacted with kapok, which is compatible with the change in Zeta potential. CTS can be simply bonded onto the negatively charged membrane, and the AuNPs with a negative charge can be immobilized onto the CTS-coated KF or CKF. Similarly, the surface of the KF/CTS/AuNPs and CKF/CTS/AuNPs becomes rough, and many imperfections are detected (Figure 13f,g), indicating that the CTS l was generated on the exterior of the KF. The AuNPs can be seen in the TEM scans of KF/CTS/AuNPs and CKF/CTS/AuNPs (Figure 13h,i) [128].

Using photocatalytic H_2_ generation inside the methanol/water solution during the simulated sun exposure, the visible light photocatalytic performance of the generated materials with varying TiO_2_ source concentrations was examined. Figure 14a,b demonstrate the photocatalytic hydrogen production during the 6-h exposure period and the hydrogen evolution level for every material. Under specified conditions, all examined samples exhibit visible hydrogen production from the photocatalytic activity. Because it can only be activated by UV exposure, a pure TiO_2_ with a wide band gap value had the lowest photocatalytic H_2_ production amount of 36.7 μmol h^−1^ g^−1^. However, based on its potential to absorb photons from the visible light area, the photocatalytic hydrogen evolution rate of pure g-C_3_N_4_ is 75.8 μmol h^−1^ g^−1^. Nevertheless, due to substantial photoinduced electron recombination, its photocatalytic hydrogen evolution rate is significantly lower when compared to other samples. In contrast, the synthesized pure CCN demonstrated a photocatalytic hydrogen evolution rate of 216.8 μmol h^−1^ g^−1^, which is approximately a three- and six-fold enhancement above pure g-C_3_N_4_ and TiO_2_, correspondingly. This enhancement was attributable to carbon doping in the CCN, which increased charge carrier separation substantially. The photocatalytic reaction durability of CCN/T-1.5 samples in photocatalytic H_2_ production was assessed by recycling the sample under comparable reaction circumstances. Following photocatalytic hydrogen evolution, the residual photocatalyst powders were collected. As shown in Figure 14c,d, there is no detectable decrease in the photocatalytic performance of H_2_ generation of sample CCN/T-1.5 after four cycles and 24 h of processing. Therefore, it was proposed that CCN/T-1.5 displayed excellent consistency for photocatalytic H_2_ generation. The outstanding consistency is critical for utilizing solar energy in operational applications. Figure 14 depicts the band gap structure of Carbon and Nitrogen co-loaded Titanium dioxide with the visible mid-gap level to demonstrate the influence of Carbon and Nitrogen loading in Titanium dioxide. As shown in Figure 14, the band gap arrangement of Carbon with g-C_3_N_4_ and Titanium dioxide crossed the redox performance for water splitting. As a result, the charge segregation mechanism of sample CCN/T-1.5 for photocatalytic hydrogen production can be represented in two ways: type II-heterojunction (Figure 14e) and direct Z-scheme (Figure 14f). In the case of the type-II heterojunction process (Figure 14e), electron excitation from VB to CB occurs in both Carbon with g-C_3_N_4_ and Titanium dioxide during photon energy absorption. Since the CB energy band of CCN is more negative than the CB energy band of TiO_2_, photogenerated electron transport from Carbon with g-C_3_N_4_ to Titanium dioxide occurs via the developed core-shell heterojunction framework. With the presence of Pt nanoparticles as cocatalysts, the deposited high-density photogenerated electrons at TiO_2_ CB concurrently interact with proton or water to produce H_2_. As a result, the photo-induced charge carriers are efficiently separated in space. Meanwhile, since TiO_2_ has a higher VB band energy level than CCN, the photogenerated h^+^ from Titanium dioxide moves to the CCN valence band. At this point, the sacrificial agent is methanol, which consumes the high-density collected photogenerated positive hole. This process may also inhibit positive hole and photogenerated electron recombination at CB of CCN [123].

The catalytic function, as depicted in Figure 15a, did not generate any significant variation in the absorbance of the aqueous solution and no H_2_ was formed in the total lack of the catalyst, demonstrating that BH^−4^ ions could not lower the CR color. The absorbance was moderately reduced after the addition of KF or KF/AuNPs to the solution, but there was no discernible decolorization of the CR color. This demonstrates that KF or KF/AuNPs have no further catalytic effect in decreasing CR color and that the only effect is decreased adsorption of KF or KF/AuNPs for CR dye. Once the KF/CTS/AuNPs and CKF/CTS/AuNPs nano-structure were applied to the CR solution, including BH^−4^ ions, the absorbance immediately dropped to near zero, and the color of the solution practically lost color during three minutes (Figure 15a). The appropriate curves of ln(C/C_0_) versus t are shown in Figure 15b, and the value constant K_obs_ (first-order rate constant) may be derived. The catalytic degradation activity constants for KF/CTS/AuNPs are 0.0171 s^−1^, 0.0229 s^−1^ for CKF/CTS/AuNPs, 0.0223 s^−1^ for CKF/CTS/AuNPs following ultrasonic treatment, and 0.0225 s^−1^ for CKF/CTS/AuNPs after acid treatment. The elimination level of CR solution catalyzed by CKF/CTS/AuNPs is higher than that of KF/CTS/AuNPs, showing that increasing the quantity of Au^0^ charged in the nanocomposite is beneficial to improve the catalytic activity (Figure 15c,d). Furthermore, after 20 min of ultrasonication and 1 mol/L HCl solution treatment, the nanocomposite retains significant catalytic interaction (Figure 15e,f), and the catalytic removal yield values of CKF/CTS/AuNPs are marginally reduced from 0.0229 s^−1^ to 0.0223 s^−1^ (after sonication) and 0.0225 s^−1^. This also shows that the nano-structure has acceptable consistency and high performance.

Figure 16a–c depicts the water-splitting hydrogen evolution outcomes from the manufactured photocatalysts exposed to visible light. The hydrogen generation on g-C_3_N_4_ increases considerably following the KF alteration. At 550, 600, and 650 °C, the H_2_ production rate of bare CN is 0.28, 0.97, and 4.06 μmol h^−1^, correspondingly. When the carbonization temperature rises by 50 °C, the quantity of H_2_ produced quadruples. The ideal hydrogen production rate for Carbon-g-C_3_N_4_ generated at various temperatures is 3.32, 6.38, and 18.89 μmol h^−1^ for 5Carbon-g-C_3_N_4_550, 2.5Carbon-g-C_3_N_4_600, and 1.5Carbon-g-C_3_N_4_-650, respectively. As a result, during visible light irradiation, the optimized sample 1.5CCN650 produced 67.5 times more H_2_ than the CN at 550 °C. The use of KF as a feedstock for CN modification is not a single factor [124,129]. According to the journal, scientists investigated the effect of pretreatment KF on the catalytic performance of CN. We also modified the KF that had been processed. Furthermore, we combined CN with paraffin, the primary element of surface biomass wax. In the KF mechanism, two factors are cellulose and paraffin. In this system, the effects of cellulose and paraffin on catalytic performance are in Nash equilibrium. Since the cellulose content is in the positive loading band and the paraffin is in the general impact level, the consequent overall KF impact is medium. In another case, both the cellulose and paraffin amounts are found in the general influence, with a moderate overall influence on catalysis. Conversely, cellulose and paraffin can be found in the positive loading band, which optimizes the general influence on catalysis and vice versa. The reported non-Gaussian distributions in Figure 16a–c are thought to be the consequence of a combination of two influencing variables. As a result, the Game Theory Nash equilibrium adequately accounts for the spontaneous fluctuation in hydrogen production rate due to KF concentration.

Figure 16e shows that the absorbance of the CR at 350 nm migrated to 360 nm, while the absorbance peak at 540 nm vanished, implying that the azo CR molecules decreased to create a new substance. Figure 16d depicts the suggested catalytic removal mechanism, which shows that the AuNPs initiate the catalytic decrease by transmitting electrons from the BH^−4^ donor to the acceptor molecules of the CR dye. In contrast, the KF/CTS/AuNPs or CKF/CTS/AuNPs can more successfully obtain and transmit electrons to the dyes [130,131]. Consequently, the azo double -N=N- bonds were minimized as the -N-N- bonds were increased, and the red CR dye disappeared (Figure 16e). The application of the nano-structure as a catalyst is critical to practical usage. Figure 16f depicts the removal rate of the CR solution after 10 cycles of color removal. It can be noticed that the color removal rate has not decreased significantly, and 92.6% of the original dye removal rate was attained following nine recoveries. This suggests that the nanocomposite could be used as a reusable catalytic substance for dye removal.

In summary, inserting kapok fiber-based carbon loading in the semiconductor structure using a simple bio-template synthesis technique can improve photocatalytic hydrogen generation in pure semiconductors. Furthermore, a well-constructed mesoporous micro-tubular structure was produced during the fabrication of a C-doped semiconductor employing treated kapok fiber. The generated C-doped semiconductor’s band gap structure and quantity of carbon doping may be easily modified using various impregnation processes. It should be highlighted that the significant increase in photocatalytic performance was due to improved light absorption, an acceptable energy band gap, and quick photogenerated carrier transfer and separation. Developing the C-doped semiconductor-based heterojunction photocatalyst in conjunction with other semiconductor materials can also aid in creating a highly excellent photocatalyst.

### 5.2. Photocatalytic Degradation

Several years ago, the elimination of color was accomplished through adsorbents depending on electron-hole pair contact. However, changes in the shape and texture of photocatalytic components have a favorable impact on the characteristics, such as surface area and photon carrying capacity, which is accompanied by the mobilization of electrons and holes along with the shape [132,133]. Several studies have proven the multiple photocatalytic abilities of several photocatalytic substances for the degradation of various organic molecules available in water, particularly colors, up to the present day [134,135,136,137]. All of these research findings on photocatalytic applications for contaminant degradation show that the bandgap, electron-hole recombination rate, size and shape, crystallinity, phase composition, light infiltration through photocatalytic substances, surface area, and dye adsorption potential on the surface of photocatalysts are important criteria for photocatalytic operation improvement. [138]. Depending on this, scientists have become interested in producing higher surface area photocatalysts, which can offer a larger surface area and more significant dye adsorption on the interface of these compounds.

Metal sulfide-based semiconductors are the most notable photocatalysts among the highly regarded photocatalysts for the removal or dissolution of colors in wastewater applications with minimal expenses, environmentally beneficial, and durable treatment solutions for environmental preservation. In previous decades, environmental contamination has become a severe hazard to the ecosystem and public safety. To combat pollution, loaded and heterojunction-based semiconductor metal sulfide nanostructures (MSNSs) are being explored as photocatalysts for photocatalytic removal or to eliminate massive industrial colors in an environmentally benign and durable approach. In the 1970s, the photocatalytic processing of water splitting to produce hydrogen on semiconductor nanostructures was discovered. The project will then discover the underlying mechanisms that result in photocatalysis and increase the system of photocatalytic removal performance [139]. The bandgap, surface area, quantity of catalyst, and formation of an electron-hole pair are all critical parameters in the photocatalytic removal of hazardous chemicals in an aqueous medium. It has been discovered that, among all parameters, the surface area has a crucial impact on the photocatalytic removal of colors by giving a larger surface area, which results in the increase of dye particle adsorption on the membrane of the photocatalyst and increases photocatalytic effectiveness. Based on the capacity to solve energy and environmental challenges, heterogeneous photocatalysis utilizing semiconductors has gained a lot of interest in recent years as an environmentally friendly and durable solution. Depending on heterogeneous photocatalysis, the analysis of the latest advancements in the synthesizing and usage of semiconductor MSNSs as photocatalysts in the field of heterogeneous photocatalytic removal of multiple colorings by ranging diverse settings like the size of the components, bandgap, light intensity, surface area, and dye solution concentration levels; and their relations with aquatic pollutants [140].

#### 5.2.1. Mechanism of Photodegradation

Photocatalytic processes can occur either homogeneously or heterogeneously. Still, recently, heterogeneous photocatalysis has received significantly more attention due to its significant value in a range of environmental and power-related purposes and molecular compound production. The photocatalytic reaction in heterogeneous photocatalysis presupposes the establishment of an interaction between a solid photocatalyst and another material holding the reagents and products of the process. As a result, the term “heterogeneous photocatalysis” is mostly applied when a light-absorbing photocatalyst is within the interaction with both a solvent and a vapor state. It has drawn continuous investigation since its inception, as seen by the large volume of decent papers, reviews, and books dedicated by different scholars to diverse heterogeneous photocatalysis processes [141]. Heterogeneous photocatalysis has proven to be an effective method for decomposing organic pollutants in the atmosphere and water, including colors, insecticides, and hazardous chemical compounds. It employs UV light in the existence of a semiconductor photocatalyst to speed up the removal of eco-system pollutants and the degradation of highly dangerous compounds [142]. However, photocatalytic degradation events take place in both homogeneous and heterogeneous environments. Solid semiconductors are generally more widely utilized because they are less expensive, more resilient, and easier to recycle and utilize than liquid photocatalysts. Considering the potential of heterogeneous photocatalysis, in contrast, homogeneous systems should not be neglected, and they may occupy a more significant function in the long term [143]. Photocatalysis can also occur in the homogeneous process by employing particles as photocatalysts because molecule-simulated steps are more significant reductants than the initial state, resulting in an electron distribution procedure. On the other hand, the composition of excited states may facilitate atom migration. As a result, colors exhibit intense bands in the UV range and the observable spectrum in homogeneous photocatalysis. Figure 17 depicts the hypothesized mechanism of photocatalytic degradation of organic dyes utilizing semiconductor MSNSs under varied light irradiations. In summary, as a dual-function modifier, KF offered a carbon supply for carbon loading and converted it into an activated carbon platform after carbonization, allowing g-C_3_N_4_ to develop on it. In addition, due to the decrease in bang gap intensity, extra electron-hole pairs are generated while the KF-altered catalyst is activated by visible light. Because of the high contact between the functional carbon substrate and g-C_3_N_4_, charges can travel quickly between them, resulting in a lower electron-hole recombination yield. As seen in Figure 17, the photoelectrons and holes are transformed into powerful oxidizing species, which oxidize the phenol to carbon dioxide and water [144].

#### 5.2.2. Kapok-Based Composite for Degradation

The catalysts were characterized by XRD, UV–Vis DRS, FT-IR, TEM, XPS, N_2_ adsorption, and PL (Photoluminescence). KF, as a double enhancer, not only furnished a carbon supply for carbon doping and converted it into an active carbon material following the carbonization process, allowing g-C_3_N_4_ to develop on it. Kapok altered graphitic carbon nitride has a more extensive surface area, finer layers, more remarkable photophysical ability, a significant separation rate, and photocatalytic degradation capacity. The phenol removal efficiency consistent with the kapok-altered graphitic carbon nitride is 0.258 h^−1^, which is 4.2 times greater than that of the neat Carbon nitride and has outstanding catalytic durability and high integrity. High carbonization frequency causes a low amount of graphitic carbon nitride in the catalyst, decreasing photocatalytic efficiency [144].

Via hydrothermal bonding with EDA and PVA, an elastic KF/GO hydrogel was created from NaClO_2_-pretreated short kapok fibers and GO sheets. After freeze-drying, the KF/GO hydrogel, a black cylinder aerogel, was formed. The as-prepared aerogel has high mechanical stability, with a standing loading of 200 g. In the calcination procedure, GO can be decreased by adding kapok fibers comprising cellulose at high temperatures, as detailed below. The free-standing CKF/RGO aerogel is mechanically strong and has a low volume (8.5 mg cm^–3^). Dog-tail grass could be used to maintain the lightweight CKF/RGO aerogel (Figure 18a). A slight decline in EDA could result in the GO sheet’s self-assembling. The KF and GO were then cross-linked into the hydrogel with minor quantity contraction due to hydrogen bonds and physical interaction between cellulose molecules and GO and the bonding impacts of PVA and GO’s large specific surface area (Figure 18b) [145].

The electrical status of surface components was investigated using XPS wavelengths. The XPS bands of CN (600) and KF (5%)-CN (600) in the area of C 1s are shown in Figure 19e,f. For CN, three peaks are found at 284.6 eV, 285.9 eV, and 288.2 eV. (600). They are related to the contaminating carbon feedstock, the final C-NH_x_ shape, and the aromatic ring’s N=C–N structure in that order. Organic emptiness kapok fibers with minimal value were used as a substrate to manufacture extremely effective polyaniline (PANI)-based adsorbents for heavy metal and chemical color contaminants. The PANI with kapok nano-structure was created through the polymerization of aniline monomer in an acidic environment with ammonium persulfate (APS) as the oxidizing component. The deposition, structure, hydrophilic behavior, and immobilization ability of the kapok fiber with PANI nano-structure were customized by executing NaClO_2_ processing and adjusting the APS to the aniline amount. The processing with NaClO_2_ was discovered to improve aniline monomer adherence on the membrane of KF, resulting in a high-performance coating. To eliminate the lignin component from the kapok fibers, NaClO_2_ was utilized as a preparatory step (KpF). In an acidic environment, NaClO_2_ creates chlorine dioxide, which causes lignin to oxidize and be removed [79,146]. The degradation rate improves the crystalline portion of the cellulosic substrate, promoting aniline infiltration and adherence [147]. The weight of KpF was lowered from 1.5 to 1.3 g following the NaClO_2_ recovery. This 13.3% diminution falls within the observed variation of lignin in KF (13–21.5%) [148]. Furthermore, the reported color shift from yellowish to white, as illustrated in Figure 19a,b, suggests that the substance is primarily composed of cellulosic components [149]. The hydrophobicity of NaClO_2_-KpF decreased as the constant water interaction position decreased from 155 to 140°.

The SEM scans of pure kapok fibers (KF) and NaClO_2_ with Kapok Fiber are presented in Figure 19g–j. Because of its hydrophobic waxy covering, the pure KF has a hollow tubular shape with a rather smooth interface. The general shape did not modify after NaClO_2_-treatment, demonstrating that there was no structural disruption. The nano-structure with (APS)/(aniline) = 1.4, on the other hand, had the maximum adsorption rate. Serial adsorption studies were used to evaluate the impacts of adsorbent amount, starting solution pH, interaction time, starting dye and heavy metal amounts, and temperature. Immobilization of MO and Pb (II) onto the Kapok Fiber with PANI nano-structure correlates strongly with the pseudo-second-order kinetic model, according to kinetic analysis. Equilibrium isotherm investigations reveal that the adsorption of each type of contaminant approaches the Langmuir isotherm model, with predicted monolayer adsorption rates of 136.75 and 63.60 mg/g for MO and Pb (II), correspondingly. Thermo-dynamic investigations show that the immobilization of MO/Pb (II) is endothermic and spontaneous [150].

Yian Zheng et al. (2012) used aniline in situ rapid polymerizations (AN) to generate an adsorbent from Kapok fiber (KF) and polyaniline (PAN). The findings reveal that KF may control the position of PAN growth, resulting in the formation of kapok Fiber with PAN nanowires serving as an adsorbent in the recovery of hexavalent chromium. [146]. The effects of operational parameters such as pH, interaction length, Chromium (VI) amount, and heavy metal coexistence were examined. The adsorption efficiency and ability were calculated using three adsorption isotherms, including the Langmuir, Redlich-Peterson, and Freundlich equations. The analysis reveals that at low starting Chromium (VI) ratios, the as-fabricated kapok fiber with PAN has a comparable immobilization capability as PAN. Because of its intrinsic broad lumen, KF/PAN has a faster adsorption capacity. Heavy metal coexistence has no visible effect on adsorption performance, concluding that PAN and KF is a very effective and cost-effective adsorbent for targeted Chromium (VI) reduction. The absorption peak at 1557 cm^−1^ in the FTIR spectrum of PAN is connected with C=N extending the Quinonoid band resonance, while 1470 cm^−1^ is related to the C=C extending resonance of a benzenoid band. Therefore, the occurrence of immobilization zones with similar strengths at 1557 cm^−1^ and 1470 cm^−1^ indicates the existence of PAN in its 50% naturally basic oxidized phase. The absorption peaks at 1290 cm^−1^ and 1239 cm^−1^ are due to the C-N extending resonance of the second amino group of the PAN structure, correspondingly, and the bipolar composition connected to the C-N extending resonance. The absorption region at 1107 cm^−1^ is caused by the 1,4-ring of C-H in-plane bends. The FTIR spectra of KF/PAN are equivalent to that of PAN, except for many unique absorption bands caused by KF. Carbonyl bonds are connected with the three major ester bands at 1741, 1374, and 1244 cm^−1^ in the FTIR spectra of KF (C=O ester). The absorption spectra at 1056 cm^−1^ are in the polysaccharide area. These natural polymer absorption ranges emerge in the FTIR spectra of Kapok with PAN or combine with the PAN absorption range, increasing absorption strengths [146].

Based on Zheng et al.’s 2012 study, kapok fiber-based polyaniline (KF-O-PAN) was produced using simple aniline polymerization on the KF interface and optimized using reaction surface approaches focusing on a centralized hybrid structure. Three sulfonated colors were used as typical adsorbates to explore the adsorption characteristics of KF-O-PAN. The effects of interaction duration, pH, preliminary color amount, and poly (vinyl alcohol) scaling on the proportion of dye adsorbed were studied using a batch test. Furthermore, the ratio of dye adsorbed in various water sources was determined [151]. To investigate the influence of ionic strength, filtered water, normal water, drill water, and lakeside water were tested. According to the findings, adsorption stability can be achieved in Four hours. The mono-layer adsorption potential determined from the Langmuir model is 188.7, 40.82, and 192.3 mg/g for OG-II, CR, and OG-G, respectively. Except for CR, KF-O-PAN exhibits a greater volume of dye adsorbed when ion intensity is considered at pH 8. The improved desorption efficacy using NaOH solution and repeated adsorption-desorption operations suggests that KF-O-PAN has the potential to remove sulfonated dyes from aqueous solutions successfully [147].

A simple surfactant-assisted approach was used by Agcaoili et al. in 2017 to create polyacrylonitrile (PAN)-coated vacuous kapok microtubes. CTAB (cetyltrimethylammonium bromide) was used to aid in polymer formation and organization. Following PAN coating, the kapok fibers maintained their hollow microtube morphology. The water interaction angle was also reduced dramatically from 133.43 to 0°. As a result, at greater CTAB concentrations, the PAN coating became denser and extra homogeneous. The insertion of a more significant proportion of PAN, on the other hand, resulted in a denser PAN coating. The impact of interaction time and temperature on MO dye and Copper (II) adsorption effectiveness from a watery medium was investigated using batch tests. According to isotherm studies, the recovery of methylene orange and Copper (II) follows the Langmuir concept, with maximal retention capacities of 34.72 and 90.09 mg/g, correspondingly. This shows that Copper (II) is trapped on KF-PAN more selectively than MO. The adsorption of methylene orange (MO) and Copper (II) on KF-PAN follows a pseudo-second-order kinetic system, identifying chemisorption as the rate-determining phase, according to kinetic analysis. Furthermore, thermodynamic statistics show that Copper (II) and MO immobilization are unexpected/endo-thermic [104].

The surface of kapok fiber is waxy and exceedingly hydrophobic. The water interaction angle of pure kapok is around 133.43°. The optical scans in Figure 20a show the impact of raising the quantity throughout PAN formation. As shown in Figure 20a, wrapping KF with PAN in the addition of 10 mg CTAB resulted in a dramatic drop to 12.15° for the interaction angle of DI water. Even at low CTAB concentrations, the KF-PAN shows an extraordinary enhancement in hydrophilic characteristics. The zero-interaction angle, on the other hand, could be attributed to the kapok membrane’s irregular loading of PAN. As shown in Figure 20a, increasing the CTAB content to 20–40 mg leads to the absence of a water interaction angle [104]. Figure 20b depicts the static water contact angles determined on PANI-KpF nanocomposites synthesized with various (APS)/(aniline) ratios. The calculated water contact angles were not consistent for (APS)/(aniline) = 0.2–0.4. When (APS)/(aniline) = 0.2 and 0.4, the value varies from 20–130° and 0–40°, correspondingly. Nonetheless, those readings are less than NaClO_2_ with a KF (140°) value. This means coating the substrate of NaClO_2_ with KF using PANI, which enhances the hydrophilic nature. A shortage of oxidizing agents could induce variations in permeability throughout the polymerization process. As a result, PANI coverage was lacking in some areas of the NaClO_2_-KpF surface. A non-angle interaction was detected by raising the (APS)/(aniline) ratio to 0.6. This suggests that a minimum of (APS)/(aniline) is required to cover the surface of the fibers thoroughly. Conversely, when the (APS)/(aniline) ratio exceeded 0.6, the water contact angle was zero [150].

The impact of UV light stimulation on the MO reduction effectiveness of the KP-ZnO-PANI nanocomposite is shown in Figure 21a. Following 60 min of interaction time, both sets of tests had eliminated approximately 20% of the dye. Furthermore, within 2 h, the nanocomposite exposed to UV radiation destroys the MO dye faster than the material in the darkness. The dye’s adsorption-desorption equilibrium with the ZnO photocatalyst was probably formed mainly around 60 min of interaction time. Further than that, UV irradiation begins to provide an impact on ZnO photodegradation. During 24 h, the KP-ZnO-PANI nanocomposite under UV light significantly destroys 10% extra methyl orange than the darker test. As a result, ZnO particles significantly improve the capacity of the KP-ZnO-PANI for water filtration [48]. COD, TOC, and total inorganic carbon (TIC) analyses were used to validate the photocatalytic degradation of RhB color. The results reveal that with the addition of Bi-TNT (Figure 21b,c), COD and TOC were reduced by 100% and 96%, respectively. TNP, Bi-TNP, TNT, and P-25 photocatalyst, on the other hand, reduced COD by 22, 25, 76, and 63%, correspondingly. Simultaneously, the percentage reduction in TOC was 20, 22, 63, and 35%. The percentage reduction in COD corresponds well with color removal data obtained from UV–visible examination. TIC analysis results were essentially steady during the response. This could be related to the mineralization of produced intermediate products into inorganic carbons. The findings clearly show that the Bi-TNT photocatalyst outperformed TNT, TNP, Bi-TNP, and P-25 in degrading RhB dye under direct solar exposure [132]. The CKF/RGO aerogel’s adsorption capabilities for different organic solvents and oils were evaluated. This aerogel demonstrated considerable adsorption capability towards a variety of organic liquids, with an adsorption capacity Q ranging from 5000 to 11 000% based on the organics’ volume, fluidity, and surface pressure (Figure 21d) [145].

KF is a sustainable resource with a high hollow lumen. Attributable to the distinct structure, low toxicity, and cost-effectiveness. The photocatalytic degradation efficiency of kapok fiber (KF) generated by the pyrolysis technique was examined. KF, as a dual-function modifier, not only provided a carbon supply for carbon doping but was also converted into an activated carbon substrate following the carbonization process, allowing the catalyst to grow on it and direct the growth orientation. All of the data suggested that the treated kapok fiber had potential uses in the sectors of wastewater treatment for dye removal, and fiber reusability assessed a possible use for metal regeneration.

### 5.3. Adsorption Using KF

Spontaneous and purposeful oil spills have emerged regularly throughout shipping, manufacturing, and refinement in recent decades, resulting in significant negative consequences for individuals and the natural system [7,143]. Oil-absorbing substances are commonly recognized as the most successful for removing and recovering wasted oil. They are classified into inorganic mineral substances, chemically synthesized polymers, and natural organic components [152,153]. Graphite, organic clay, vermiculite, silica, perlite, fly ash, and zeolites are examples of inorganic mineral minerals [154,155,156,157,158,159,160]. In recent decades, polyacrylate, polyethylene, and polypropylene, a new substance for adsorbing oil, are synthesized organic polymers [160,161,162,163]. Various agricultural items are used in organic natural resources, including cotton fiber, kenaf, straw, milkweed, sawdust, and straw kapok fiber [164,165,166]. Among these organic products, kapok offers several benefits over typical oil-absorbing materials, including inexpensiveness, renewability, innate hydrophobicity, and high sorption ability, making it desirable as an oil-absorbing substance [167]. All recent research regarding the adsorption application of kapok fiber is illustrated in Table 2.

#### 5.3.1. The Main Parameters Effect on Kapok Fiber

The toluene absorbance of kapok fiber processed with NaOH, NaClO_2_, and HCl solvent is shown in Figure 22a. The toluene absorbance of treated kapok fiber enhances with increasing concentrations of NaOH, NaClO_2_, and HCl but declines with subsequent treatment intensity rising up to 3%. The decrease in toluene absorbance is noticeable for kapok fiber treated by NaOH, which decreased ranging from 31.2 g/g to 24.3 g/g. The commonly applied solvent during pre-treatment is NaOH for pure fibers preparatory to chemical or physical alteration [75]. Wax compounds, natural oils, and pectin that coat the exposed layer of kapok fiber can be removed by NaOH processing, exposing the fibrils and resulting in a rugged texture. Equally, HCl and NaClO_2_ treatments can cause identical fluctuations, but their effects on the micro-texture of KF are considerably less than NaOH processing [172]. The hollow aperture of kapok fiber that can retain oils dissolves partially or fully during this phase, and the form gets fattened. The NaOH amount utilized in this treatment is relatively low, but the destructive impact on the capillary form is nevertheless noticeable, causing the overall oil absorbency to fall continually. On the other hand, NaClO_2_ and HCl treatments enhance the crystalline loading arrangement in kapok fiber, allowing better penetration of compounds such as toluene [173]. The bleaching of NaClO_2_, in particular, might effectively eliminate a considerable portion of the lignin and has a more enlarged matrix in fiber, facilitating solvent adsorption. The temperature has a varied consequence on the toluene absorbance of treated kapok fiber, as illustrated in Figure 22b. Toluene absorbency reduces significantly with rising temperatures for NaOH-treated fiber and suddenly with the rising temperature for HCl-treated fiber from 80 °C to 100 °C. In alkaline-processing KF, the phenomena are mainly related to the destruction of the aperture and the conversion of the airy kapok fiber organization into a condensed lignocellulose arrangement. Hydrolysis can disrupt certain H_2_ bondings and the structure of KF in the HCl treatment. In addition, HCl will partially eliminate hemicellulose and cellulose at higher temperatures, resulting in a smaller fiber length in form. As a result, toluene may be challenging to implicate inside the cylindrical lumen. In addition, NaClO_2_ processing at extreme temperatures makes it simpler to eliminate lignin and other non-cellulose elements, including xylan and waxy compounds, from kapok fiber while preserving the integrality of the entire structure. The fiber barrier of kapok fiber is primarily made of lignin, hemicellulose, and cellulose, with a tiny crystallized region formed by H_2_ bonding among these molecules [59,174]. Nonetheless, the partly crystalline zone may be converted to an unstructured area by NaClO_2_ bleaching, allowing solvents to penetrate the matrix of the aperture surface. The toluene absorbency of water-treated kapok rises with the rising temperature. The majority of studies show that the waxy kapok surface contributes significantly to the capability of adsorbing oil [60,61]. Based on Jintao Wang et al.’s 2012 investigation, top wax removal can also significantly increase toluene absorption capacity., This is due to the benefits of the hard texture following chemical processing. The crude surface improves the bonding capacity of oil in kapok fiber arrangements, and elevated temperatures could promote the formation of interface ruggedness [8]. Figure 22c depicts the connection between treatment time and toluene absorption. As previously discussed, along with the increase in treatment time, the oil absorption of NaOH and HCl-treated kapok fiber decreases dramatically. Chloroform-extracted kapok fiber shows a slightly diminishing tendency in terms of toluene absorbency. Furthermore, chloroform extraction can affect the crystalline phase of kapok fiber, allowing extended fibers to change into smaller fibers or particle states, resulting in a diminished kapok fiber arrangement’s ability to capture the oil. However, since chemical processing with NaClO_2_ maintains kapok fiber’s robust and porous shape, prolonging the treatment period does not affect toluene absorption. The findings suggest that the hydrophilic kapok fiber arrangement retains superior oil-capturing capabilities. The oil coating that protects the exterior interface of the cellular fiber is not the solitary component that controls the flow of oil across the kapok fiber’s extra-lumen arrangement. As a result, we may determine that, with proper processing, kapok fiber is a significant lignocellulosic resource for oil absorption in the circumstances of a lack of water [61,175]. Figure 22d depicts the toluene absorbance of treated kapok fiber throughout eight adsorption rounds. The results show that the toluene absorbance of all samples drops dramatically within the first three adsorption rounds, accompanied by a slightly steady reduction. An identical finding was found in the research of oil adsorption on kapok fiber of Malaysia [61]. The irreparable distortion of half cavity lumens, the compression of fiber inter-space, and the existence of leftover oil in fiber layers all contribute to a decrease in oil absorbance [153]. Considering this, the toluene absorbance of kapok fiber processed with HCl, NaClO_2_, and water is consistently more significant than raw kapok in all measurements. As a result, chemical processing for kapok fiber may be regenerated numerous cycles without losing performance, showing its potential as an oil-capturing material for collecting various oils via basic pressing.

#### 5.3.2. Kapok Fiber Characterization as Adsorbent

Balela et al., 2019 used KF-PAN nanocomposites with progressive quantities of AN monomer to conduct serial adsorption tests for Pb (II) ions. The greatest immobilization for Pb (II) ions in the KF-PAN nanocomposite produced by 4.5 mL AN monomer was approximately 26.81 mg/g, as shown in Figure 23a. The immobilization capacities of KF-PAN nanocomposites containing 1.5 and 3.0 mL AN were approximately 22.04 and 14.70 mg/g, correspondingly. The growth in PAN thickness with AN monomer content can be related to the fluctuation in Pb (II) adsorption capability. In addition, PAN is reported to bind numerous metal ions significantly because of the polarity of nitrile atoms [105]. Due to a singular couple of electrons on the N_2_ molecule, the nitrile band serves as an H_2_ connecting receiver. It also has a substantial polarity position among the electron-defect carbon molecule and the particle N_2_ molecule [176], which allows the active group to form extraordinarily strong interparticle bonds [177]. As the PAN layer diameter increases with rising AN monomer content, extra nitrile groups may be produced, resulting in better Pb (II) removal efficiency [150].

Using kapok fibers coated with NaClO_2_, methylene blue (MB) was successfully recovered from a liquid solution. The RSM based on a three-factorial Box-Behnken design was employed throughout the adsorbent purification process. The impacts of three parameters on the MB adsorption process were investigated: NaClO_2_ quantity (0.3–1.2 g), acetic acid (HAc) volume (0.1–1.9 mL), and heating rate (60–90 °C). The results of the statistical analysis were used to determine the best treatment parameters for producing an adsorbent and extracting MB from a liquid medium. The influences of interaction duration, pH, and mechanisms on adsorption were examined. The model’s estimated amount (105.48 mg/g) agreed perfectly with the observed result (110.13 mg/g). The adsorption occurred quickly and with pseudo-second-order kinetics. The adsorbent displayed high recyclability with 0.1 mol/L of HCl as a solubilizing reagent and could be used as a suitable adsorbent for cation-colored wastewater purification. [80].

Since FTIR analysis is often utilized to determine the molecules responsible for the process mechanism, the FTIR analyses of pure and modified kapok fiber were studied and are presented in Figure 23b. It was stated that NaClO_2_ would create chlorine dioxide within the acidic environment, rendering lignin oxidation impossible. In addition, the wide spectrum at approximately 3397 cm^−1^ attributed to the bending vibration of O-H in cellulose caused the material to appear wider following processing, and this may be due to the breaking of certain H_2_ atoms and lignin. As a consequence, the abstract component of fiber increases and more hydroxyl molecules are released. The range at 1643 cm^−1^ may be related to the deflection resonance of H_2_O particles, and it did not change after processing. The absorption spectrum at 1592, 1504, and 1463 cm^−1^ practically vanished due to bending vibrations of C-C in separate replacement aromatic chains in lignin. The absorption spectra at 831 cm^−1^, due to the swinging oscillation of C-H in the 1,4-disubstituted aromatic chain in lignin, practically vanished after processing. According to most of the evidence, the treatment dissolved the lignin in the kapok fiber. Moreover, the finding may substantiate that the coated kapok fiber demonstrated excellent hydrophilicity and was utterly wet after 30 min of stirring with distilled water. Despite being swirled for 48 h, the pure kapok fiber remained in the distilled water [80].

Carbon nanotubes, such as nanofibers, have been created via the combustion of leaf tissue, which was influenced by the nature of plant tissue. However, synthesizing cylindrical structures by incineration is difficult since the form is dynamically non-steady and always breaks throughout the calcination operation. Carbon microtubes were synthesized using a basic pyrolysis process using kapok. The cylindrical shape with diameters ranging from 5 to 20 mL was generated from the resulting carbon material, demonstrating an identical form to kapok sources, but the weight was dropped by around 90%. Zhao et al., 2019 proved that carbon microtubes have outstanding oil adsorption capability, with an adsorption rate of 190 g g^−1^, 1.5 times greater than pure kapok. Furthermore, they also discovered that it might be reused more than ten times in an oil adsorption operation. Moreover, the adsorption capabilities of carbon microtubes dramatically expanded when the temperature was raised during calcination [178].

Figure 23c depicts kapok’s thermogravimetry curves. Because of moisture reduction, the weight reduced somewhat from 20 to 65 °C by around 9%. The curves then plummeted steeply from 240 to 370 °C for nearly 70% of the time, showing that the crucial chemical processes occur at this point. As the temperature was raised from 500 to 1000 °C, 13% of the mass of the raw components was converted to the final carbon elements, and the mass amount is no longer moving, indicating that no further interactions were performed in this area. The TGA studies indicate that the calcination phase happened at temperatures ranging from 240 to 370 °C. Previous research has shown that high temperatures favor carbon substances by increasing permeability and particular contact zones. [179]. As a result, we attempted to raise the calcination temperature to modify the microstructure of the tubes to enhance adsorption capacity [178]. Raman spectra were utilized to determine the G-band and D-band of carbon materials to determine the chemical compositions and crystallinity of the carbon microtube. As illustrated in Figure 23d, two noticeable peaks were discovered around 1340 cm^−1^ (D-band) and 1595 cm^−1^ (G-band), corresponding to the carbon crystallinity imperfection and the in-plane bond stretching of C–C bonds, showing the production of amorphous carbon in the fiber.

In conclusion, the modified kapok fiber performed exceptionally well in removing pollutants and oil from aqueous solutions. The adsorption capacity of KF maintained steadily throughout five rounds of adsorption-desorption studies, showing that KF may be employed as an efficient adsorbent for reducing contaminated wastewater. From both an environmental and functional approach, KF may be a viable adsorbent for detoxifying contaminated wastewater.

### 5.4. Photocatalytic CO_2_ Reduction

With global warming and resource scarcity becoming increasingly extreme, the electrochemical CO_2_ minimization process (ECO_2_RR) is an effective and intriguing strategy to convert CO_2_ into various valuation compounds as a carbon-neutral way to a sustainable power source [180,181,182]. Among the several compounds created by CO_2_ electroreduction, formate or formic acid is a significant fluid biofuel that can be effectively exploited as a super-economic power transporter in fuel cell applications [183,184,185]. Biodegradable resources (animal or botanic) are near-ideal solutions for the synthesis of several high-production catalysts, which have numerous essential advantages, such as inexpensive cost, easy availability, and molecular and geometrical variations [186,187,188]. Kapok fiber is an organic porous fiber generated from the silk-cotton tree with a significant cavity and a thinner covering and is used as an oil and heavy metal ion adsorbent [56,169]. Furthermore, kapok fiber is an ideal carbon source for the fabrication of electrochemical supercapacitors [46,129]. The use of kapok fiber in electrochemical CO_2_ removal, on the other hand, has never been recorded. The enormous cylindrical form of kapok-tubes is advantageous for carrying metal nanoparticles; also, kapok-tube has a high capacity for metal ion capturing due to its numerous oxygen active element on the membranes [98,168]. Most of the evidence suggests that it can be used as a novel carbon substrate to manufacture electrocatalysts. The kapok tube is mainly employed as a carbon electrocatalyst for Carbon dioxide electroreduction, and it performs well for electrocatalytic Carbon dioxide to fluid energy conversion without adding active components. Unlike typical carbon nanotubes and graphene, which have weak capacities for formate generation without adding active components, this natural capacity may be attributed to the many mesoporous structures found in MHKTs, which acted as functional spots for formate generation. Additionally, the kapok tube is used for the first time as a novel catalyst substrate for depositing metal nanoparticles in electrocatalytic CO_2_ removal. All the metals in-situ bonded on MHKTs are manufactured using a simple one-pot synthetic technique. For the CO_2_-to-formate process, the four electro-catalysts, Sn, Bi, Pb, and Cd-MHKTs, exhibited great selection, low overpotential, high existing density, and prolonged stability. These low-cost metal MHKT electrocatalysts offer many potential applications in electrocatalytic CO_2_ degradation to formate.

In addition, the structure of mesoporous hollow kapok tubes is studied using SEM (MHKTs). Figure 24b shows stretched kapok tubes with a consistent diameter of around 10 m. In Figure 24c, the porous form of kapok tubes is clearly visible. Furthermore, the extreme enlargement SEM data of hollow kapok in the caption of Figure 24c shows that many mesoporous appear on the kapok-tube interface, with a specific contact coverage of about 1068.3 m^2^ g^−1^ and a mesopore size of 3.64 nm. Furthermore, TEM is employed to describe the MHKTs, and multiple mesoporous are clearly visible in Figure 24e. As evaluated by EDS elemental analysis, c and O elements are uniformly dispersed in MHKTs (Figure 24f,g). Figure 24h shows an XRD pattern of MHKTs with a large, limited peak at 24.1°. Carbon has a structural spacing of 0.368 nm over (002) orientation. Furthermore, XPS-analysis is used to investigate MHKTs deeply; only Carbon and Oxygen components are clearly observed in the data set XPS findings in Figure 24i, with O having a higher density content than C, implying that MHKTs have extensively available oxygen active element on their membranes, which are preferable for capturing metal particle. Figure 24j shows high-frequency C1s spectra that can be linked to C–C, C–O, and C=O peaks with corresponding bound values of 284.8, 285.6, and 287.7 eV.

MHKTs demonstrate significant electrocatalytic activity for CO_2_ reduction. As shown in Figure 24k, the initial CO_2_ minimization possibility of MHKTs is around −0.7 V. Figure 24l depicts existing electrolysis concentrations of MHKTs over different applied potentials recorded by steady electrolysis; existing densities increase as applied potentials increase. In ordinary circumstances, to further evaluate the capacity for electrocatalytic CO_2_ reduction, electrolytic evaluations are performed at potentials ranging from −0.7 to −1.3 V for five hours in ordinary circumstances. The identified aqueous material is mainly formate at these hypothetical variations, and the vapors are only carbon dioxide and hydrogen. Figure 24m depicts the Faradaic Efficiency (FE) of formate, carbon oxide, and hydrogen collected at MHKTs. The formate FE for MHKTs shows a trend in which the formate FE gradually improves as electrolytic performance significantly boosts from −0.7 to −1.0 V, achieving the maximum formate FE at a promising area of −1.0 to −1.1 V, then declining after −1.1 V due to increasing hydrogen. As a consequence, typical carbon substances, including 2D graphitic oxide (GO) or 1D carboxyl multi-walled carbon nanotubes (MWCNTs), were used to make comparisons. Furthermore, they showed a low formate generation capacity, implying that various mesoporous elements in MHKTs functioned as operational formate generating locations. The defects generated by these numerous mesoporous surfaces by carbon dioxide processing served as productive locations for carbon dioxide removal in MHKTs. Therefore, the effect of N atom elimination can be ignored further. Tafel plot analysis is performed in Figure 24n to get a further mechanism understanding. The Tafel gradient of MHKTs is 122 mV dec-1, which is close to the hypothesized estimate of 118 mV dec-1, suggesting that the chemical rate-determining step (RDS) is the primary one-electron transmission to form the deposited carbon dioxide.

In conclusion, the usage of Kapok fiber for CO_2_ conversion under visible light to create solar fuels is discussed, as well as how various properties and structural adjustments may impact the processes and final products. The presence of a flexible structure for adjusting band gaps and imparting lattice distortion allows kapok fiber to control separation, mobility, and the lifetime of photogenerated charges. Demonstrating good selectivity for format generation across a wide range of potentials. The use of kapok fiber offers fresh insight into the production of innovative carbon supports with suitable photocatalytic characteristics.

## 6. Conclusions and Future Recommendations

### 6.1. Conclusions

Due to its natural benefits, kapok fiber has attracted more consideration as an eco-friendly resource in current decades. Despite the growing demand for kapok fiber, the difficulties in producing it restrict its production potential for a broader range of purposes. The creation of clean and compostable materials has become a center of concern as worldwide environmental consciousness has grown. KF-based or KF-derived materials have piqued the attention of researchers in materials science and environmental science due to their distinctive hollow structure and hydrophobic-lipophilic surface. KF can be merged with or supplied with other elements, bonded using the hydroxyl segments linked to the fiber, layered with active polymers containing tailored groups, or carbonized to form new active substances utilizing crosslinking methods. Due to its thin-walled porosity and hydrophobic-lipophilic interface, KF has proven to have a substantial capacity as one form of oil-capturing substance, exhibiting fast absorption kinetics, high absorption efficiency, and strong uptake specificity for the concept oils. Following that, certain specialized oil-capturing systems and purification processes were created. This review summarized several applications of kapok fiber, including adsorption, degradation, hydrogen production and CO_2_ reduction. Based on the limited studies that have been published on hydrogen production and CO_2_ reduction, many elements of kapok fiber exploration and improvement probably stay unexplored. In conclusion, KF is a biomass resource for utilization into a worthwhile material that enhances present research and spreads the fiber to developing industries such as fuel and catalysis. As a result, despite its present usage, KF has the potential to serve a much further function in upcoming studies and usage.

### 6.2. Future Recommendation

Nevertheless, most KF research has focused on unstructured fibrous formations, and additional study on structured fibers is needed to enable additional feasible uses. Considering more attention to this sustainable cellulosic fiber, more research should be conducted to extend the applicability domains for kapok fiber by exploiting its variety of great vacuousness and hydrophobic–oleophilic properties. As a significant oil substance, the majority of uses are currently restricted to minimal experimental work, and additional investigations in handling realistic ecosystem difficulties will be seriously addressed for considerable utilization. A variety of KF-based adsorbents have been developed to facilitate the upgrading of natural plant fibers by attaching the fiber layer to various external active groups. However, because of the inherent hydrophobicity of the native fiber, hydrophilic alteration is often required by substantially or entirely eliminating the wax adhered to the fiber surface. Because of its unusual thin-walled hollow tube, KF has recently gained more interest in energy-related applications. Simple hydrothermal or pyrolysis carbonization is often the main functionalization process for realizing its full potential as an electrode material for supercapacitors. Durability and recyclability should be illustrated in the current scientific study. Aside from employing green raw resources, addressing the durability of the eventual usage is a high priority. Then, KF-based materials have piqued the interest of researchers as photocatalysis and catalysis substances for ecological and power purposes. Combusted fiber is the research emphasis for catalytic materials, considering cost minimization and fiber shape for high stability as novel difficulties. The economic potential, diagnostics, and toxicity of intermediates and an entire mechanism for complete processing in environmental-related areas or improved hydrogen generation in energy-related fields should be highlighted for catalytic usage.

## Figures and Tables

**Figure 1 molecules-27-08107-f001:**
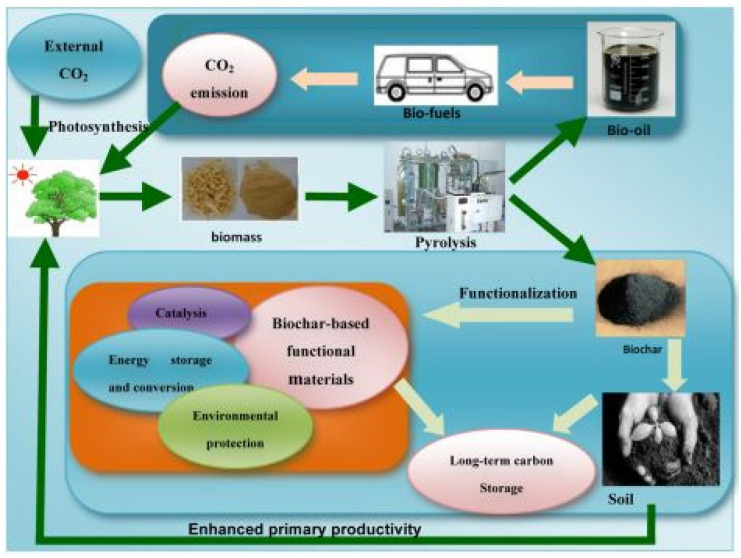
Representation of sustainable principles in biochar synthesis, uses, and overall climate effects. Reprinted with permission from Ref. [40]. Copyright 2015, copyright American Chemical Society.

**Figure 2 molecules-27-08107-f002:**
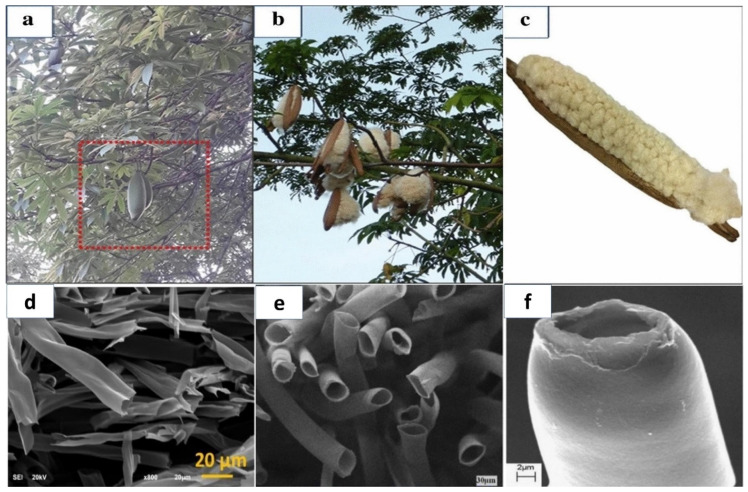
(**a**) Kapok Fiber tree, (**b**) A kapok fruit, (**c**) KF fruit. Reprinted with permission from Ref. [66]. Copyright 2017, copyright Elsevier; SEM scans represent the body (**d**). Reprinted with permission from Ref. [67]. Copyright 2021, copyright Springer Nature; hollow structure (**e**) and tail of kapok fiber (**f**). Reprinted with permission from Ref. [49]. Copyright 2015, copyright Springer Nature.

**Figure 3 molecules-27-08107-f003:**
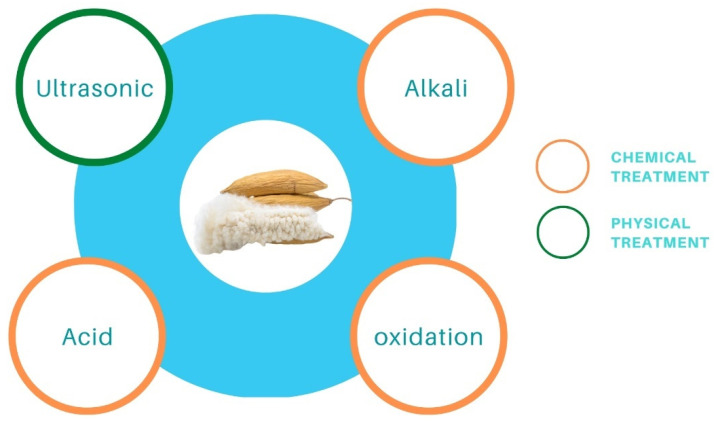
Several treatment methods for kapok fiber to remove the wax.

**Figure 4 molecules-27-08107-f004:**
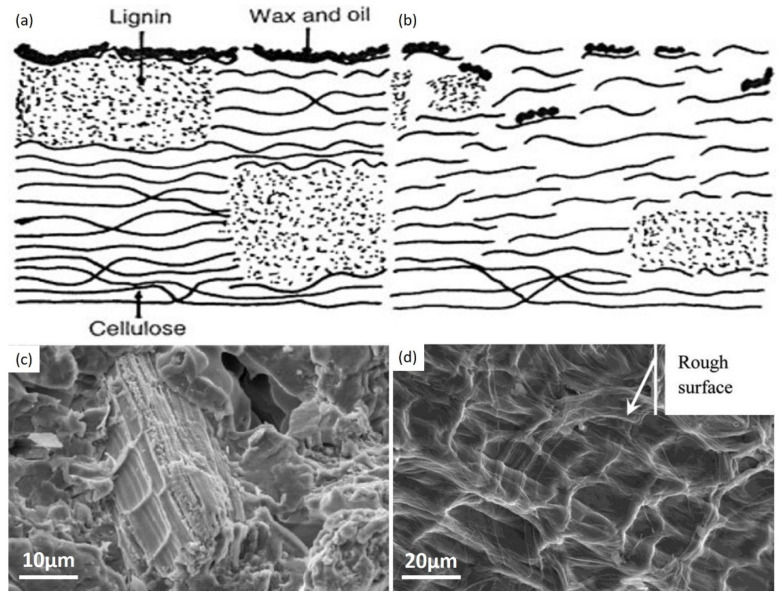
Demonstration of the difference between the pure kapok fiber (**a**–**d**) alkalized Kapok Fiber. Reprinted with permission from Ref. [75]. Copyright 2002, copyright John Wiley and Sons.

**Figure 5 molecules-27-08107-f005:**
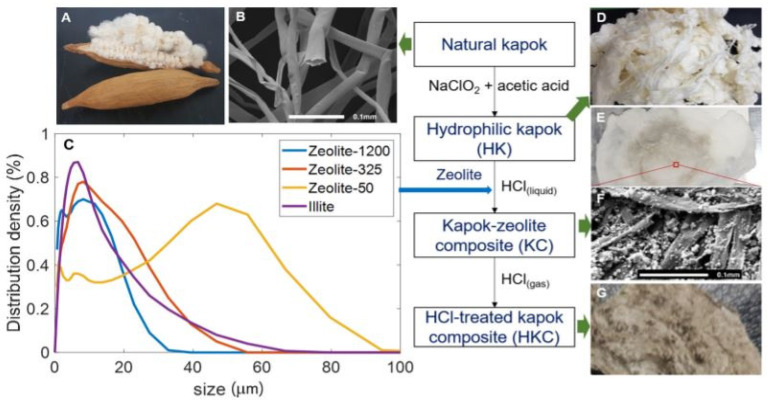
(**A**) Image. (**B**) SEM scans of natural kapok fibers. (**C**) Size dispersion of adsorbent particles as assessed by dynamic light scattering devices. (**D**) After hydrophilic pretreatment, kapok fiber becomes whiter and firmer. (**E**) A fine kapok layer surrounds the zeolite fragments, and (**F**) flattened kapok fibers are heaped without any gaps for the zeolite particles to fall through. (**G**) HCl-treated kapok fiber turns grey and gets thinner. Reprinted with permission from Ref. [76]. Copyright 2021, copyright Elsevier.

**Figure 6 molecules-27-08107-f006:**
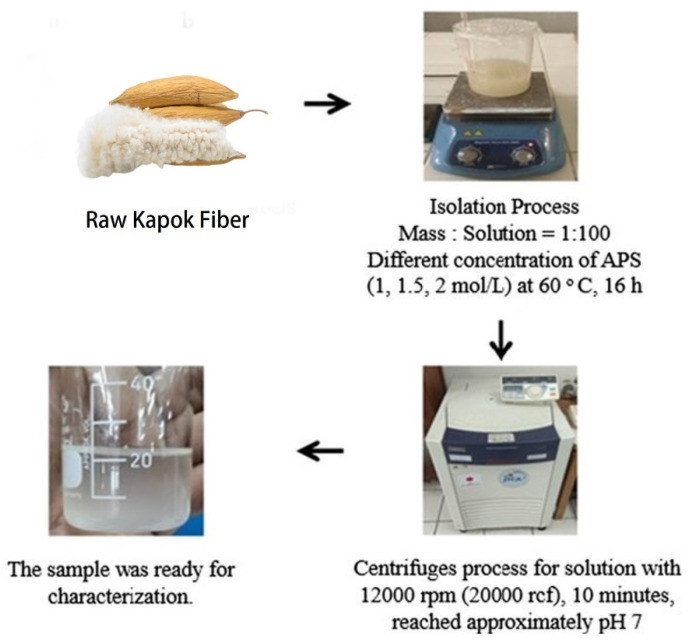
Isolation process of kapok fiber with the oxidation method. Reprinted with permission from Ref. [83]. Copyright 2021, copyright Springer Nature.

**Figure 7 molecules-27-08107-f007:**
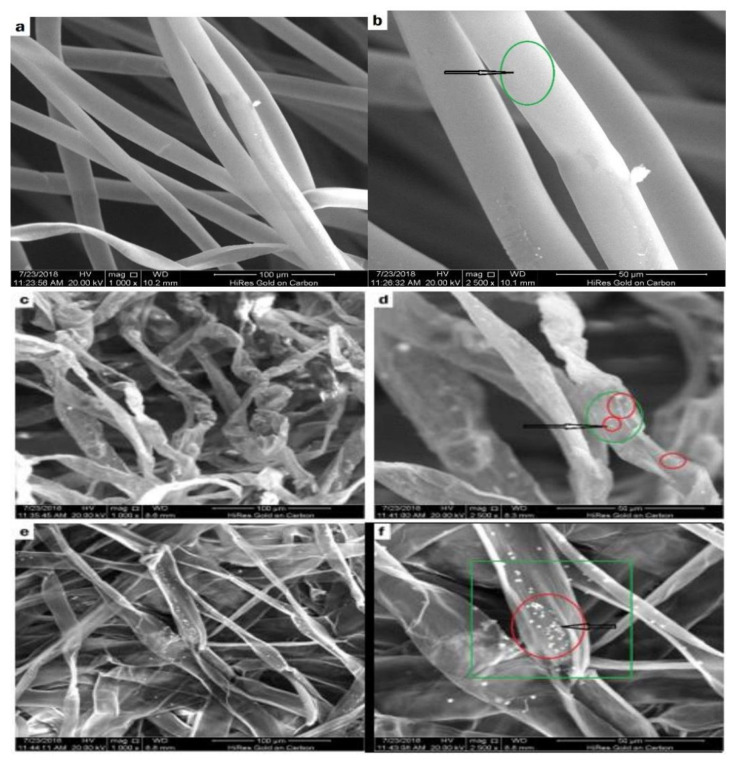
SEM images SEM Micrograph of (**a**,**b**) untreated kapok, (**c**–**f**) NaClO_2_-NaOH—treated kapok [84].

**Figure 8 molecules-27-08107-f008:**
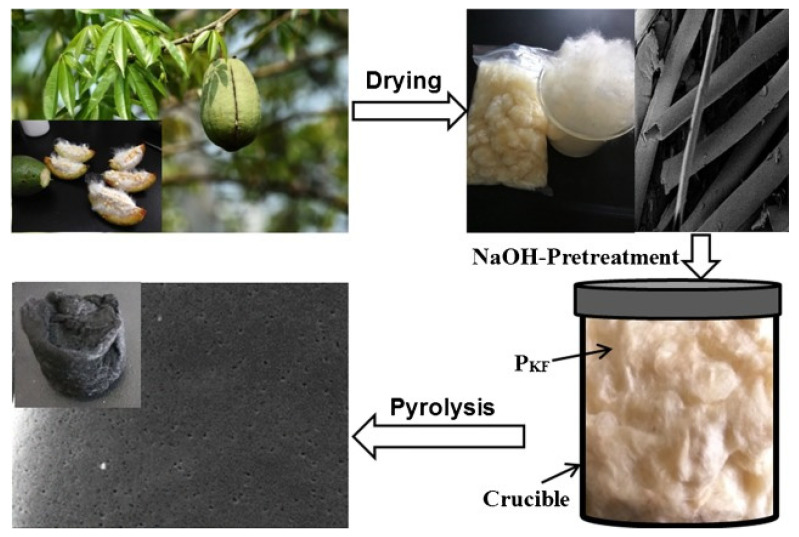
Representation of a pyrolysis Kapok Fiber processing methodology. Reprinted with permission from Ref. [90]. Copyright 2018, copyright Elsevier.

**Figure 9 molecules-27-08107-f009:**
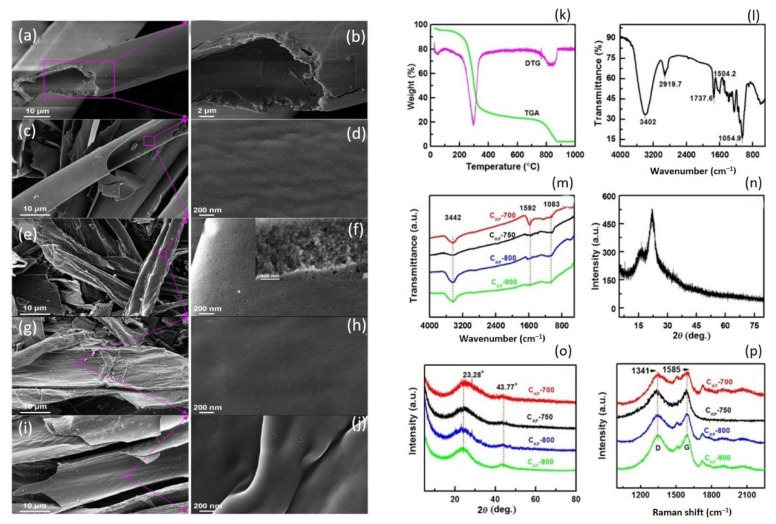
FE-SEM scans of the KF (**a**,**b**), CKF-700 (**c**,**d**), CKF-750 (**e**,**f**), CKF-800 (**g**,**h**), and CKF-900 (**i**,**j**). TGA and DTG curves of natural KF (**k**), FT-IR spectra of natural KF (**l**) and biochars (**m**), XRD patterns of natural KF (**n**) and pyrolysis KF (**o**), and (**p**) pyrolysis KF Raman spectra. Reprinted with permission from Ref. [90]. Copyright 2018, copyright Elsevier.

**Figure 10 molecules-27-08107-f010:**
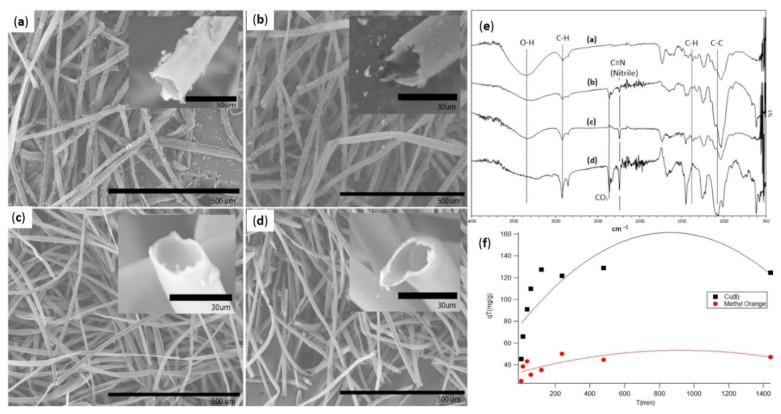
SEM scans of kapok fibers after PAN coating with (**a**) 10, (**b**) 15, (**c**) 20, and (**d**) 30, CTAB. The PAN-kapok fibers are seen with extreme intensity in the inset. Kapok fibers are 30 mg, PAN is 1.5 mL, and KPS is 60 mg; (**e**) FTIR bands of (**a**) pure kapok, (**b**) PAN-kapok mixtures made with (**a**) 10, (**b**) 20, and (**d**) 40 mg of CTAB. (**f**) The impact of interaction time on the adsorption performance of PAN-kapok hollow microtubes for Copper (II) and Methyl Orange from an aqueous solution. Reprinted with permission from Ref. [105]. Copyright 2017, copyright Elsevier.

**Figure 11 molecules-27-08107-f011:**
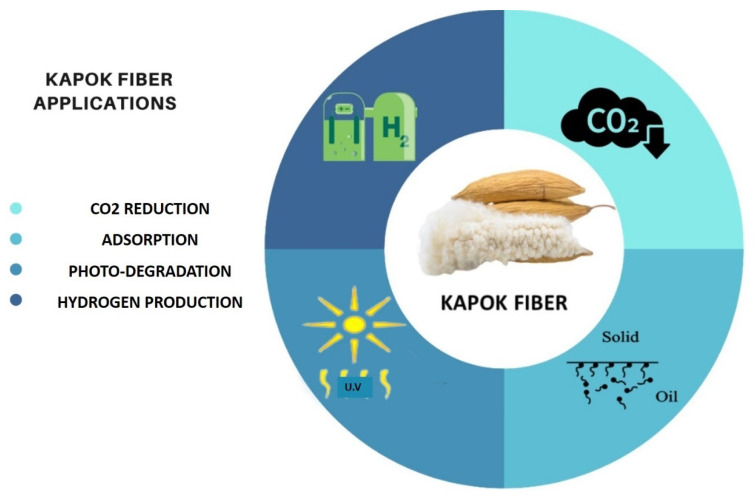
Several applications of Kapok Fiber.

**Figure 12 molecules-27-08107-f012:**
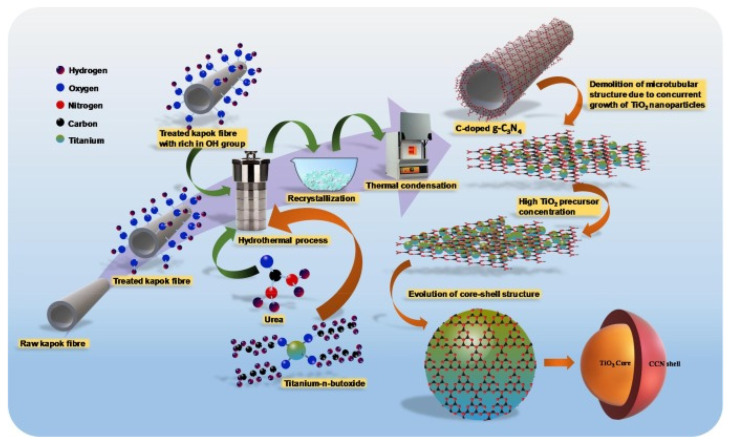
The production of KF-doped g-C_3_N_4_/C, N co-doped TiO_2_ heterojunction photocatalysts and the development of their core-shell nanostructures are depicted schematically. Reprinted with permission from Ref. [123]. Copyright 2019, copyright Elsevier.

**Figure 14 molecules-27-08107-f014:**
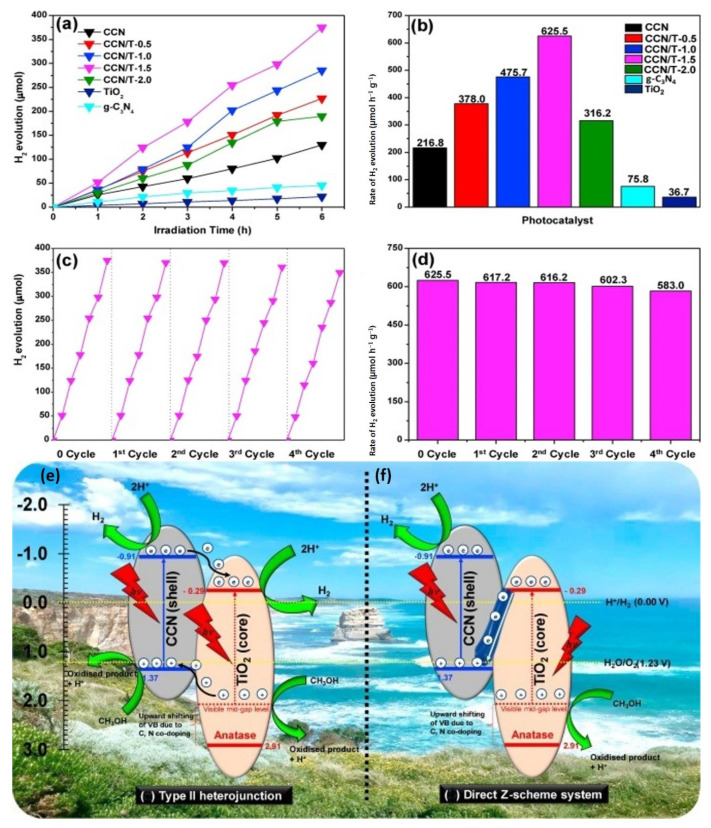
(**a**) photocatalytic hydrogen production rate; (**b**) photocatalytic hydrogen evolution efficiency comparing several photocatalysts. (**c**,**d**) photocatalytic consistency and photocatalytic H_2_ generation frequency over CCN/T-1.5 for four cycles. During simulated solar exposure, a schematic representation of the photocatalytic process of CCN/T-1.5 photocatalysts is shown. Type II heterojunction (**e**) and Direct Z-scheme system (**f**). Reprinted with permission from Ref. [123]. Copyright 2019, copyright Elsevier.

**Figure 15 molecules-27-08107-f015:**
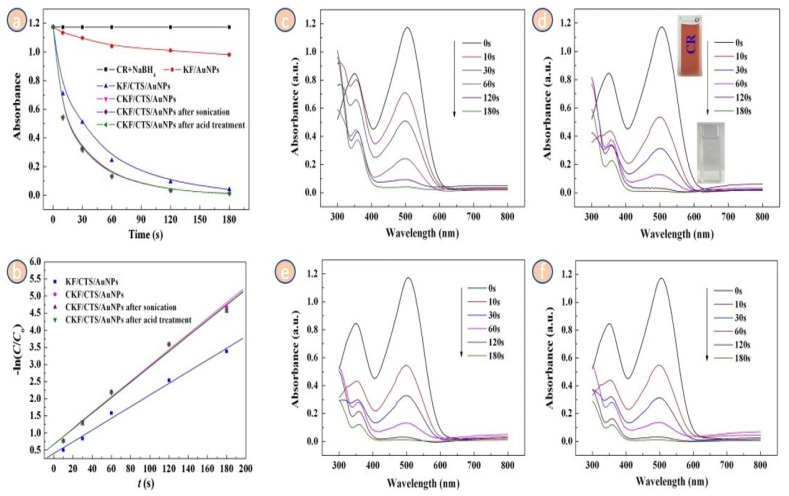
(**a**) KF/AuNPs, KF/CTS/AuNPs, and CKF/CTS/AuNPs catalytic kinetic profiles for CR process degradation; (**b**) linear correlation of ln(C/C0) per time t; (**c**)Time-dependent UV–vis absorption wavelengths for the catalytic removal of CR by NaBH_4_ in the existence of KF/CTS/AuNPs, CKF/CTS/AuNPs before (**d**), during, and after sonication (**e**), and after acid treatment with 1 mol/L HCl solution (**f**). Reprinted with permission from Ref. [128]. Copyright 2014, copyright Elsevier.

**Figure 16 molecules-27-08107-f016:**
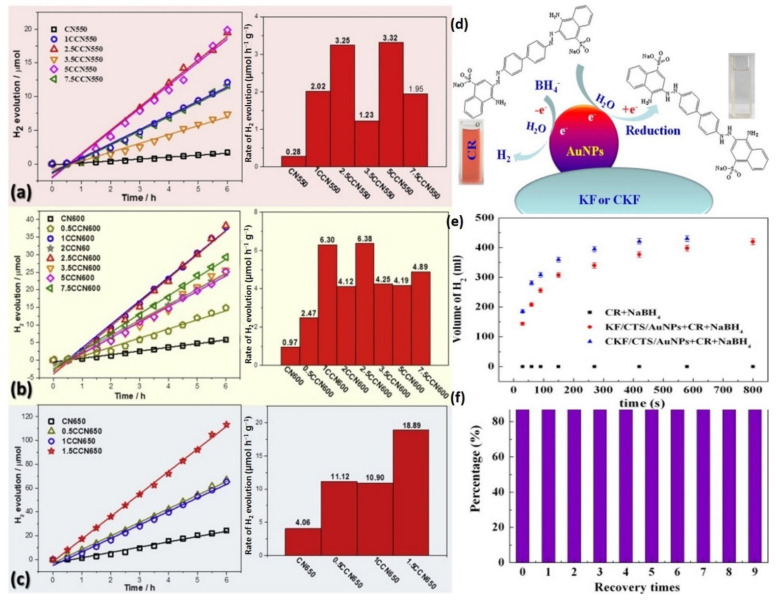
(**a**–**c**) Photocatalytic hydrogen production of g-C_3_N_4_ and kapok altered g-C_3_N_4_ composite created at various temperatures under visible-light exposure in 6 h and the associated Hydrogen production level. Reprinted with permission from Ref. [125]. Copyright 2019, copyright Elsevier; (**d**) process for catalytic removal of CR dye by NaBH_4_ employing KF/CTS/AuNPs or CKF/CTS/AuNPs as catalysts are presented. (**e**) The proportion of hydrogen generated varies over time; (**f**) After 10 cycles, the catalytic productivity of the CKF/CTS/AuNPs nanocomposite. Reprinted with permission from Ref. [128]. Copyright 2014, copyright Elsevier.

**Figure 17 molecules-27-08107-f017:**
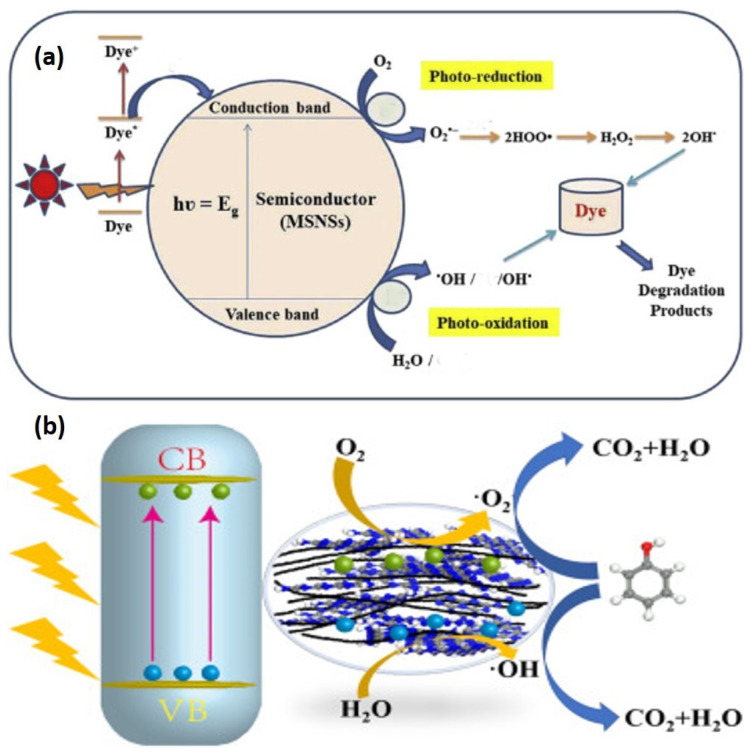
(**a**) Schematically depicts the photocatalytic removal detailed process of colors employing the exposed semiconductor metallic sulfide nanoparticles. Reprinted with permission from Ref. [140]. Copyright 2018, copyright Elsevier; (**b**) The photocatalytic degradation pathway of KF-modified g-C_3_N_4_ is depicted schematically. Reprinted with permission from Ref. [144]. Copyright 2020, copyright Elsevier.

**Figure 18 molecules-27-08107-f018:**
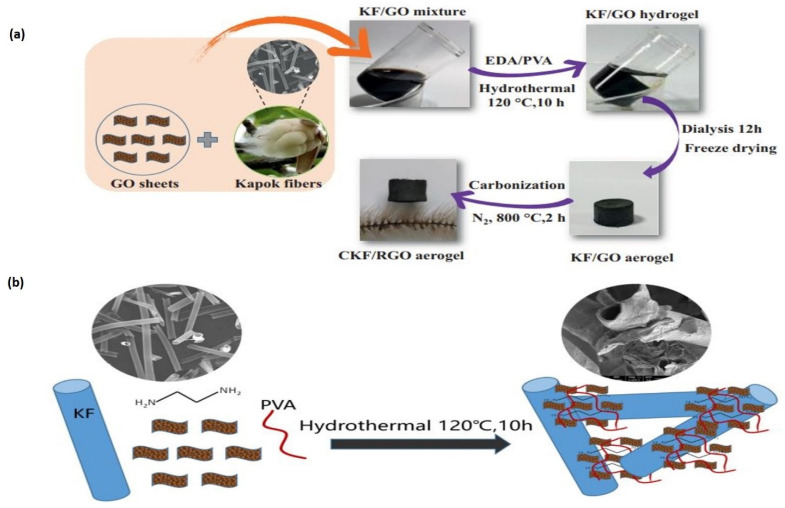
(**a**) An experimental flow diagram for preparing a CKF/RGO aerogel. (**b**) The KF/GO aerogel production mechanism. Reprinted with permission from Ref. [145]. Copyright 2020, copyright John Wiley and Sons.

**Figure 19 molecules-27-08107-f019:**
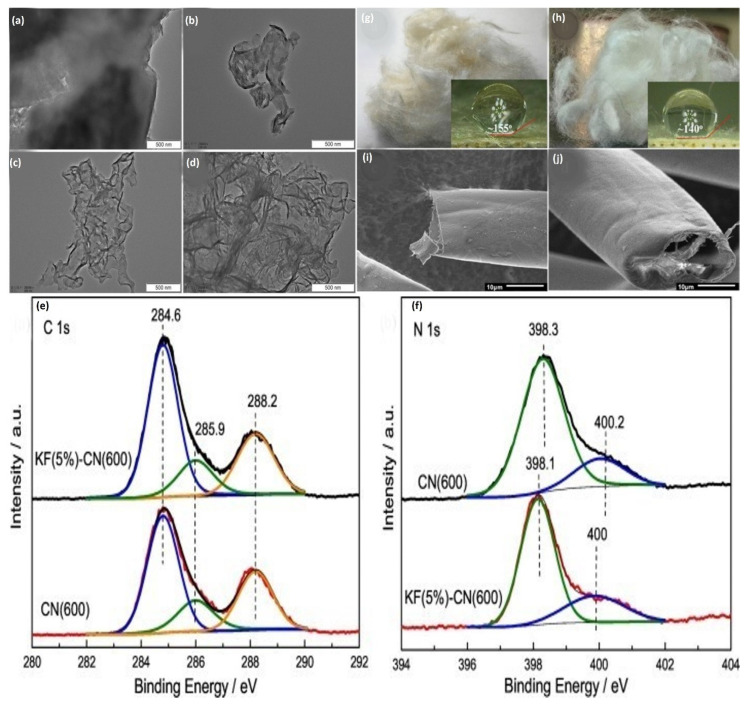
TEM scans of g-C_3_N_4_(600) (**a**), Kapok (1%) with g-C_3_N_4_(600) (**b**), Kapok (5%) with g-C_3_N_4_(600) (**c**), and Kapok (10%) with g-C_3_N_4_(600) (**d**). The XPS spectrum of (**e**) KF (5%)-CN (600) and (**f**) CN (600). Reprinted with permission from Ref. [144]. Copyright 2020, copyright Elsevier; Visual and scanning electron microscopy (SEM) pictures of pure kapok fibers (KpF) (**g**,**h**) and NaClO_2_-KpF (**i,j**) with matching stable water interaction angle values. Reprinted with permission from Ref. [150]. Copyright 2020, copyright Elsevier.

**Figure 20 molecules-27-08107-f020:**
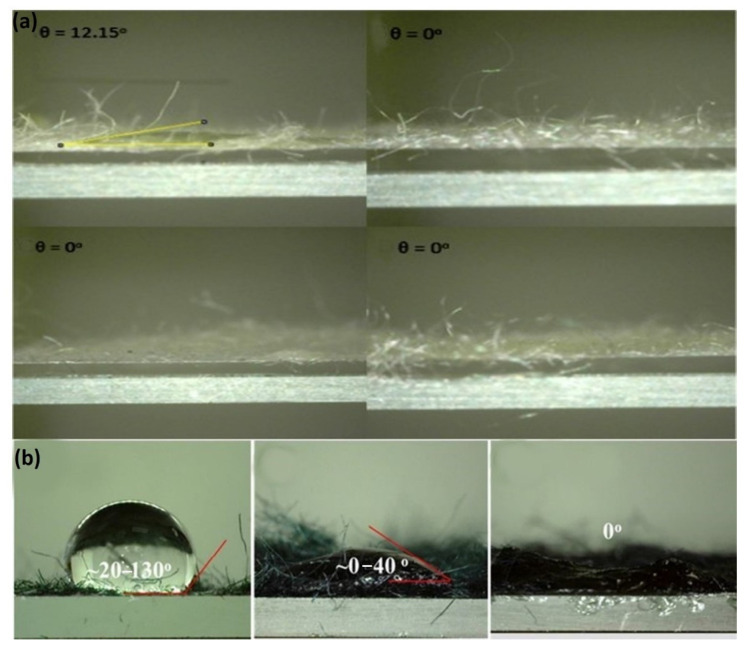
(**a**) DI water interaction angle with PAN-KF made with 1.5 mL AN, 10, 20, 30, and 40 mg of CTAB. Reprinted with permission from Ref. [105]. Copyright 2017, copyright Elsevier; (**b**) PANI-KpF static water contact angle created using (APS)/(aniline). Reprinted with permission from Ref. [150]. Copyright 2020, copyright Elsevier.

**Figure 21 molecules-27-08107-f021:**
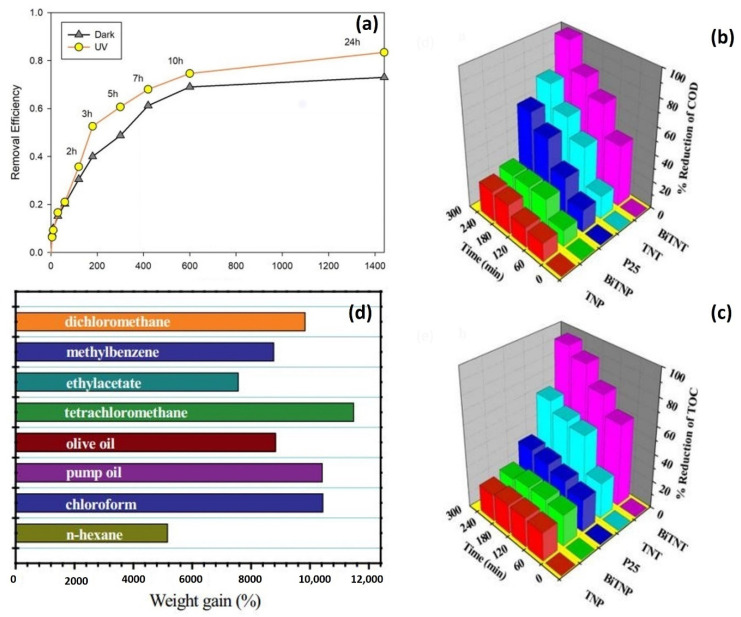
(**a**) The absorption rate of MO (25 mg/L) employing KP-ZnO-PANI in the dark and under UV light. The graphic shows the dye’s UV–Vis spectra at different time ranges under. Reprinted with permission from Ref. [48]. Copyright 2018, copyright Elsevier; (**b**) Reduction in COD and (**c**) TOC values as a percentage. Reprinted with permission from Ref. [133]. Copyright 2013, copyright Springer Nature; (**d**) CKD/RGO aerogel adsorption capability towards various organic solvents and oils. Reprinted with permission from Ref. [145]. Copyright 2020, copyright John Wiley and Sons.

**Figure 22 molecules-27-08107-f022:**
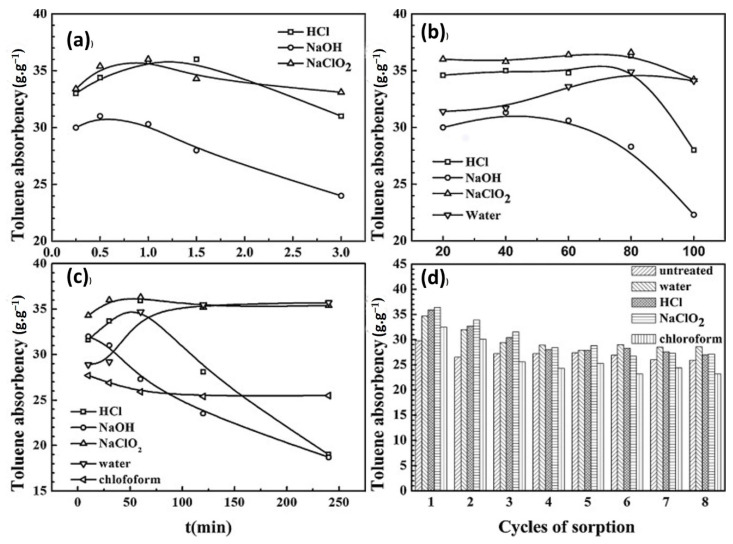
(**a**) The impact of treatment intensity on the absorbance of oil (T° = 20 °C; t = 1 h). (**b**) The impact of treatment temperature on oil absorption is as follows: NaOH, NaClO_2_, and HCl solution concentrations are 0.25%, 1%, and 1%, correspondingly; duration is 1 h. (**c**) The impact of the procedure period on oil absorption properties is as follows: the concentrations of NaOH, NaClO_2_, and HCl solutions are 0.25%, 1%, and 1%, correspondingly. The temperature is 80 °C. (**d**) Crude and processed kapok fiber durability. Reprinted with permission from Ref. [8]. Copyright 2012, copyright Elsevier.

**Figure 23 molecules-27-08107-f023:**
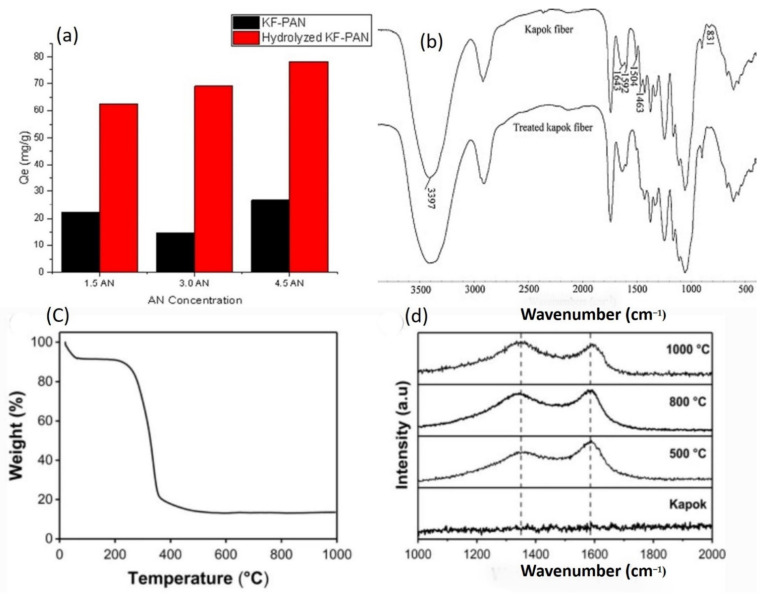
(**a**) Adsorption potential of Purified KF-PAN Nanocomposites made of KF-PAN at 200 mg/L Pb(II) for 24 h utilizing 30 mg of adsorbent. Reprinted with permission from Ref. [168]. Copyright 2019, copyright Elsevier; (**b**) Kapok fiber FTIR analysis prior/post NaClO_2_ application. Reprinted with permission from Ref. [80]. Copyright 2012, copyright Elsevier; (**c**) The kapok TGA curve. (**d**) The Raman wavelengths of kapok and carbon microtubes that carbonized at temperatures of 500 °C, 800 °C, and 1000 °C, correspondingly. Reprinted with permission from Ref. [178]. Copyright 2019, copyright Springer Nature.

**Figure 24 molecules-27-08107-f024:**
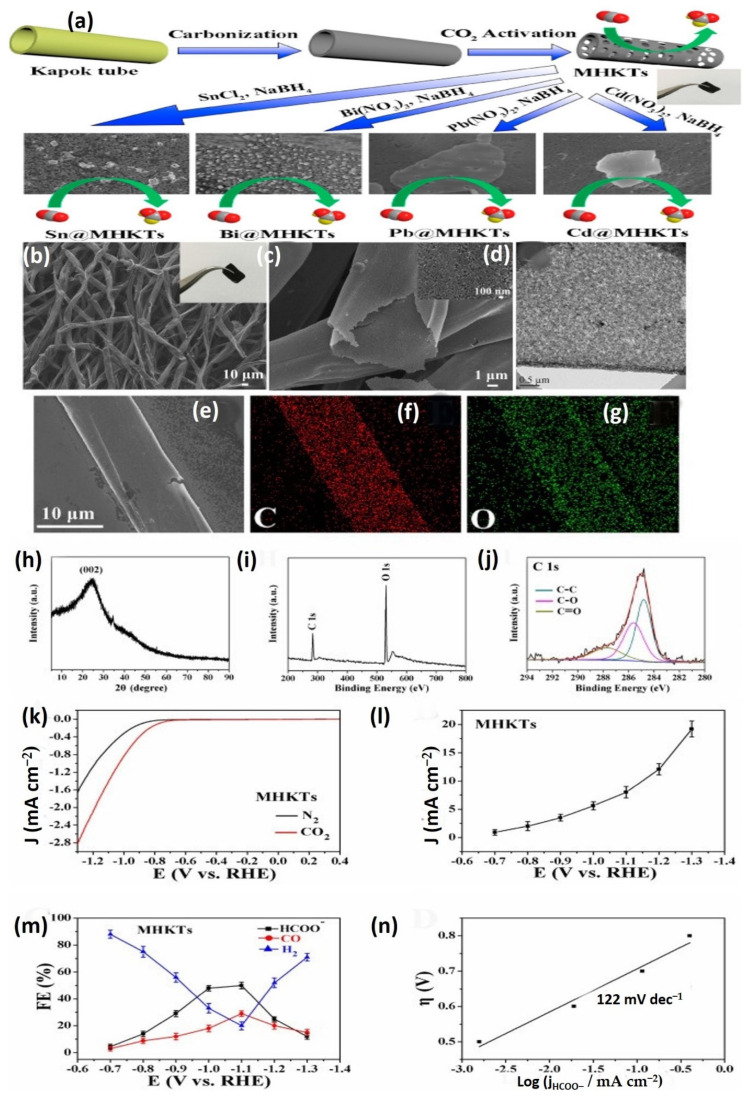
(**a**) The synthesis of MHKTs, Sn, Bi, Pb, and Cd attached to MHKTs and the usage in electrocatalytic carbon dioxide transforming into the format, is represented as a schematic; (**b**–**d**) SEM scans of MHKTs in low and high micrographs. (**e**) MHKT TEM scan. (**f**,**g**) EDS elemental analysis of MHKTs. (**h**) XRD pattern. (**i**) Scan XPS spectra. (**j**) High-definition C1s spectrum of MHKTs. (**k**) MHKT LSV in 0.5 mol dm3 KHCO_3_ product overloaded Carbon dioxide and Nitrogen, scanning frequency 20 mV s1. (**l**) Present electrolytic intensity with applied electrolysis potential variation for five-hour Carbon dioxide electrolysis at MHKTs. (**m**) Faradaic performances for five-hour Carbon dioxide electrolysis at MHKTs against different electrolysis potentials. (**n**) Tafel plot for MHKTs. Reprinted with permission from Ref. [88]. Copyright 2019, copyright Elsevier.

**Table 1 molecules-27-08107-t001:** Kapok Fiber Chemical Composition.

Composition	Percentage (%)	
Cellulose	35	[17]
Hemicellulose	-	
Lignin	21.5	
Holocellulose	84	
Ash	1.05	
Wax	2.34	
Moisture	11.23	
Xylan	22	
Acetyl Group	-	

**Table 2 molecules-27-08107-t002:** Summary of adsorption application of Kapok Fiber.

Materials	Pollutants	Adsorption Results	References
Polyaniline-kapok fiber-nanocomposite	Anionic-methyl-orange	136.75 mg/g	[146]
Kapok fiber	Methylene blue	110.13 mg/g	[80]
Polyaniline-kapok fiber-nanocomposite	Lead ions	78.34 mg/g	[168]
Polyacrylonitrile-coated-kapok hollow-microtube	methyl-orange & Cu (II) ions	34.72/90.09 mg/g	[105]
Kapok fiber-oriented polyaniline	Sulfonated dyes	192.3 mg/g	[147]
Kapok fiber-oriented polyaniline-nanofiber	Cu (II) ions	145.54 mg/g	[147]
Polyaniline-coated kapok fiber	Methyl-orange & copper (II) ions	81.04 mg/g	[10]
Hydrophilic modified kapok fiber	Lead(II)	94.41 mg/g	[14]
Acetylated modification kapok fiber	Oil	84.4 g/g	[81]
Oxidized kapok fiber	Pb, Cu, Cd and Zn	93.55%, 91.83%, 89.75% and 92.85%	[69]
Kapok fiber-based carbon microtube aerogel	Oil/organic solvents	98% (distillation) 97% (Squeezing) 90% (Combustion)	[89]
DTPA-modified kapok fiber	Pb^+2^, Cd^+2^, Cu^+2^	310.6 mg/g, 163.7 mg/g, 101 mg/g	[169]
Kapok fiber	Diesel	45 g/g	[77]
Kapok fiber	Oil	32.31 g/g	[170]
Raw kapok fiber/pyridine-catalyzed kapok Fiber/NBS-catalyzed kapok fiber	Diesel	30.5 g/g 36.7 g/g 34 g/g	[56]
PBMA/SiO_2_	Diesel, Soybean oil, Crude oil, 150SN, 20CST	99.7%, 65%, 41.1%, 23.1% and 26.8%	[58]
PBMA-Kapok Fiber	Toluene and chloroform	14.6 g/g and 26 g/g	[57]
Superhydrophobic—Kapok Fiber	Diesel and Soybean oil	46.9 g/g and 58.8	[55]
Kapok Fiber—Dopamine	Mercury	235.7 mg/g	[171]

## Data Availability

No data was used for the research described in the article.

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
