# Peer review of "Biobased Kapok Fiber Nano-Structure for Energy and Environment Application: A Critical Review"

_molecules, 2022, doi:10.3390/molecules27228107_

Round 1

Reviewer 1 Report

1. Title lengthy and not reflected to main content of the study.

2.  Abstract lack information. Background study, problem statement, points out research gaps, aims/objectives, summary of methods, and novelty of research study are not highlight clearly.

3. Introduction. The gap between previous study and current review are not clearly presented.

4. Every subsection must be include the new contribution from authors. Not only conclusions from review articles.

Author Response

The manuscript has been revised by considering all the comments and suggestions provided by the reviewer. The response to every comment has been summarized in a Table as attached.

Reviewer 2 Report

The authors could have given general information regarding importance of sustainable materials and why they are relavant in todays world.

Also authors should discuss the importance and uses of biochar in general, discuss different ways to get biochars, then can come to Kapok Fiber biochar.

Author Response

(The authors gave the same response as above.)

Round 2

Reviewer 1 Report

The revised manuscript is acceptable.